# Nucleation–mode particle pool and large increases in $N_{cn}$ and $N_{ccn}$ observed over the northwestern Pacific Ocean in the spring of 2014

Juntao Wang[1], Yanjie Shen[1], Kai Li[2], Yang Gao[1, 3*], Huiwang Gao[1, 3], Xiaohong Yao[1, 3*]

[1] Key Lab of Marine Environmental Science and Ecology, Ministry of Education, Ocean University of China, Qingdao 266100, China
[2] National Marine Environmental Forecasting Center, Beijing, 100081, China
[3] Laboratory for Marine Ecology and Environmental Science, Qingdao National Laboratory for Marine Science and Technology, Qingdao 266071, China

**Correspondence:** Yang Gao (Yanggao@ouc.edu.cn) and Xiaohong Yao (xhyao@ouc.edu.cn)

**Abstract.** Determination of the updated concentrations of atmospheric particles ($N_{cn}$) and the concentrations of cloud condensation nuclei ($N_{ccn}$) over the northwestern Pacific Ocean (NWPO) are important to accurately evaluate the influence of aerosol outflow from the Asian continent on the climate by considering the rapid changes in emissions of air pollutants therein. However, field observations in the last two decades are scarce. We conducted a cruise campaign over the NWPO to simultaneously measure $N_{cn}$, $N_{ccn}$ and the size distribution of aerosol particles from day of year (DOY) 81 to DOY 108 of 2014. The mean values of $N_{ccn}$ at supersaturation (SS) of levels 0.2% and 0.4% were $0.68 \pm 0.38 \times 10^3$ cm$^{-3}$ and $1.1 \pm 0.67 \times 10^3$ cm$^{-3}$, respectively, with an average of $2.8 \pm 1.0 \times 10^3$ cm$^{-3}$ for $N_{cn}$ during the cruise over the NWPO. All are approximately one order of magnitude larger than spring observations made during the preceding two decades in the remote marine atmosphere. The larger values, against the marine natural background reported in the literature, imply an overwhelming contribution from continental inputs. The calculated activity ratios (ARs) of the cloud condensation nuclei (CCN) were $0.30 \pm 0.11$ and $0.46 \pm 0.19$ at SS levels of 0.2% and 0.4%, respectively, which are almost the same as those of upwind semi-urban sites. High $N_{ccn}$ and CCN activities were observed from DOY 98 to DOY 102, when the oceanic zone received even stronger continental input. Excluding biomass burning (BB) and dust aerosols, good correlation between $N_{ccn}$ at 0.4% SS and the number concentrations of >60 nm particles ($N_{>60 \, nm}$) was obtained during the entire cruise period, with a slope of 0.98 and $R^2 = 0.94$, was obtained, and the corresponding effective hygroscopicity parameter ($\kappa$) was estimated to be 0.40. A bimodal size distribution pattern of the particle number concentration was generally observed during the entire campaign when the $N_{>90 \, nm}$ varied largely. However, the $N_{<30 \, nm}$, accounting for approximately one-third of the total number concentration, varied narrowly, and two NPF events associated with vertical transport were observed. This implies that a pool of nucleation–mode atmospheric particles is aloft. BB and dust events were observed over the NWPO, but their aerosol contributions to $N_{cn}$ and $N_{ccn}$ were minor (i.e., 10% or less) on the monthly time scale.

## 1 Introduction

In the atmosphere, aerosol particles known as condensation nuclei (CN) activated as cloud condensation nuclei (CCN) can indirectly affect the climate by altering clouds and precipitation (Yu and Luo, 2009; Yu et al., 2013). The role of marine atmospheric particles in the global climate system has attracted significant amounts of attention because the oceans cover nearly two-thirds of the Earth's surface; such research has identified a feedback mechanism between the ocean and climate as the Charlson, Lovelock, Andreae, and Warren (CLAW) hypothesis (Charlson et al., 1987; Quinn and Bates, 2011). In marine atmospheres, aerosol particles can be derived from a variety of sources such as sea spray aerosols, long-range transport aerosols, and the secondary aerosols formed via marine gaseous precursors or mixed marine and continent-derived gaseous precursors (Clark et al., 1998; Andreae and Rosenfeld, 2008; Feng et al., 2012; Liu et al., 2014; Guo et al., 2016; Feng et al., 2017, Hu et al., 2018). For example, ultrafine sea-salt aerosol particles released locally can be an important source of CCN in the marine atmosphere (Clarke et al., 2006). Ocean-derived dimethylsulfide and other volatile organics can undergo photochemical reactions, and their oxidized products may nucleate into particles that subsequently increase in size and act as CCN (Andreae and Rosenfeld, 2008; Hoffmann et al., 2016). The seasonal outflows of continental primary aerosols, such as dust particles and biomass burning (BB) aerosols, can also influence the number concentration of cloud condensation nuclei ($N_{ccn}$) in the remote marine atmosphere (Nagao et al., 1999; Huebert et al., 2003). The chemical and physical properties of marine aerosols thus have strong variation owing to the dynamically varying contributions from these primary sources and secondary formation pathways; therefore the $N_{ccn}$ and the CCN activity of aerosols depend strongly on maritime locations and seasons (Clark et al., 1998; Roberts et al., 2006; Lana et al., 2012; Collins et al., 2013; Dunne et al., 2014). Owing to practical difficulties, direct observation of $N_{ccn}$ in the remote marine atmosphere is limited. This prevents reliable estimates of $N_{ccn}$ over oceans, leading to larger error in the results of aerosol–cloud interaction estimates (Rosenfeld et al., 2019; Sato and Suzuki, 2019).

The northwestern Pacific Ocean (NWPO) reportedly receives large amounts of continental anthropogenic air pollutants, BB aerosols, natural dust, and volcanic plumes in the winter and spring seasons, carried by the East Asian Monsoon (Uematsu et al., 2004; Guo et al., 2016; Luo et al., 2016; Yu et al., 2016). The Kuroshio Extension causes the NWPO to be an active subtropical cyclone basin (Hu et al., 2018). Oceanic cyclones in this region enhance the primary production (Chang et al., 2017); thus air–sea exchange has a potential influence on the atmospheric chemistry (Quinn et al., 2015). Interactions between moisture and atmospheric aerosols have been proposed to deeply affect the climate (Deng et al., 2014; Wang et al., 2014). However, direct observational data of particle number concentration ($N_{cn}$) and $N_{ccn}$ remain limited in the remote atmosphere over the NWPO; the last spring observation can be traced back to 1996 (Nagao et al., 1999). On the other contrary, dynamic changes in emissions of anthropogenic pollutants have occurred on the Asian continent during the last two decades (Chan and Yao, 2008; Liu et al., 2013). Updated studies are thereby urgently needed to characterize the atmospheric particles and activation of aerosol particles as CCN therein, such as $N_{cn}$ and $N_{ccn}$, particle number size distributions, particle sources, Kappa value and activation diameter. Moreover, modeling studies show that the NWPO likely experienced the largest increases

in surface sea temperature and $CO_2$ sink under the warming climate (John et al., 2015; Lauvset et al., 2017). This further demonstrates the importance of researching $N_{cn}$ and $N_{ccn}$ and the related potential climate effects therein.

In this work, we measured the particle number size distributions, $N_{ccn}$, and other gaseous and particle pollutants in the marine atmosphere during a campaign from 18 March to 22 April 2014, or day of year (DOY) 77 to DOY 112 2014 on the research vessel (R/V) *Dong Fang Hong 2* traveling from the marginal seas of China to the NWPO and back to remedy the data scarcity of the last decades. We study the spatiotemporal variability of $N_{cn}$ and particle number size distributions, as well as the $N_{ccn}$ and CCN activities of aerosol particles with particular attention on measurements made from DOY 81 to DOY 108, when the R/V traveled in the NWPO. Through comprehensive comparison with observations in the literature, we illustrate the characteristics of $N_{cn}$ and $N_{ccn}$ over the NWPO in 2014 and reveal their changes against the results measured two decades ago. In addition, the influences of dust and BB aerosols on $N_{cn}$ and $N_{ccn}$ are analyzed on the monthly time scale. This work may lead to additional studies to update the influence of the current outflow from the Asian continent on the climate.

## 2 Experimental design

### 2.1 Field observation

In this study, we conducted measurements of atmospheric particles during a cruise campaign on the R/V *Dong Fang Hong 2* over the NWPO during DOY 77–112 in 2014 (Fig. 1 and Fig. S1). All instruments were placed in the lab at the sixth floor of the vessel approximately 15 m above sea level. Atmospheric particles were sampled through conductive tubes (TSI, US) connected with a diffusion dryer filled with silica gel (TSI, US) and a splitter that split the air flow into different instruments. The tube inlet was stretched from the window of the cabin linking to the bridge. The total sampling line was approximately 1.5 m and the loss for particles > 10 nm is tested to be negligible. The number size distributions of aerosol particles with mobility diameters of 5.6–560 nm with 19 size bins covering the size range below 100 nm and 13 size bins beyond 100 nm were measured using a high time resolution (1 s) particle sizer, i.e., a Fast Mobility Particle Sizer (FMPS, TSI Model 3091). The high time resolution enabled the FMPS to effectively screen out the signal of the ship's self-emissions (Yao et al., 2005; Liu et al., 2014). Although the particle size reported by the FMPS showed errors against the results measured by the scanning mobility particle sizer (SMPS) (Lee et al., 2013), the errors were reasonably corrected in this study by using the empirical correction procedure proposed by Zimmerman et al. (2015). Owing to a malfunction of the Condensation Particle Counter (CPC, TSI Model 3775) during the campaign, a coefficient of 1.25 was used to correct the total particle number concentration measured by the FMPS. This coefficient was reasonably stable for side-by-side measurements between the FMPS and a CPC during several previous and subsequent campaigns, as shown in Fig. S2. A continuous flow CCN counter (CCNC, DMT Model 100) was used to measure the bulk CCN concentration. The total flow rate of the CCNC was set to 0.45 L min$^{-1}$, with a sheath to sample flow ratio of 10. The CCNC was calibrated with ammonium sulfate particles following the procedure of Rose et al. (2008) prior to our observations. In this experiment, the $N_{ccn}$ at five different supersaturation (SS) levels of 0.2%, 0.4%, 0.6%, 0.8%, and 1.0% were measured. Collection of $N_{ccn}$ at each SS level required 5 min; however, the

first one minute of data was discarded to establish SS equilibrium. Completing a cycle from 0.2% SS to 1.0% SS required 28 min, including an additional 3 min from 1.0% of SS to 0.2% of SS to ensure a steady state. Owing to rough sea conditions, the FMPS was out of service from DOY 87 to DOY 95. The CCNC apparently performed well during the entire observational period according to additional measurements made after the campaign.

In addition, 19 total suspended particle (TSP) samples were collected during the campaign. The sampling duration varied from ~10 h to 22 h, depending on concentrations and anchoring times. The BB tracers were analyzed by using a gas chromatography mass spectroscopy (GC–MS); details of the procedures used for analysis of the organic tracers appear in Feng et al. (2015). Moreover, sulfur dioxide ($SO_2$) was measured by a $SO_2$ Analyzer (Model 43i, Thermo Fisher Scientific), nitrogen oxides ($NO_x$) were measured by a $NO_x$ Analyzer (Model 42i, Thermo Fisher Scientific) and ozone ($O_3$) was measured by an

$O_3$ Analyzer (Model 49i, Thermo Fisher Scientific) in 5–min time resolution. The detection limits of the gas analyzers were 0.5–1 ppb and zero drifts were 1-2 ppb throughout the campaign. It should be noted that the measured mixing ratios of $SO_2$ and $NO_X$ during the cruise campaign apparently incurred large analytic errors at level of 1–2 ppb and below. These data were thus used only for identifying the plumes from the ship's self–emissions and were not used for characterizing background levels of $SO_2$ and $NO_x$ in the remote marine atmosphere.

**2.2 Web-based data and on-site meteorological data**

The three–day air mass back trajectories were calculated by using the Hybrid Single-Particle Lagrangian Integrated Trajectory (HYSPLIT) model from the National Oceanic and Atmospheric Administration (NOAA) Air Resources Laboratory to identify the origins of air masses. The three–day backward trajectories starting at 1000 above mean sea level (a.m.s.l.) were performed every 4 h. The fire spots that occurred in East Asia and the Siberian region were detected by the Fire Information

for Resource Management System (FIRMS) using the Terra and Aqua satellites (http://firefly.geog.umd.edu/firemap). Meteorological data, including wind speed and direction, temperature, pressure, and relative humidity were measured simultaneously on board the R/V.

**2.3 Data screening**

To study the aerosol particles derived from local marine emissions and long-range transport from continents, we first

carefully screened out the data with errors from the ship's self-emissions. We exhaustively removed the suspected data that conformed to the following conditions: (1) the particle number concentration increased dramatically in a short time accompanied by an increase in the mixing ratios of pollutant gases; (2) the median mobility diameter of the dominant particle mode in those concentrated plumes was 22 ± 2 nm; and (3) the wind speed was ≤1 m/s. Although this process could lead to removal of some signals other than those intended, the remaining measurement signals effectively represented the actual

concentrations of $N_{cn}$ and $N_{ccn}$ in the marine atmosphere. An example is given in (Fig. S3) in the Supporting Information. After screening out the ship's self-emission signals, approximately 50% of all measurements were available for data analysis.

# 3 Results and discussion

## 3.1 Spatiotemporal variation of $N_{cn}$ and $N_{ccn}$

Fig. 2 shows the time series of the 1 min–averaged $N_{cn}$, bulk $N_{ccn}$, and CCN activity ratio (AR) at SS values of 0.2% and 0.4% during the measurement from DOY 77 to DOY 112, 2014. A rapid decrease in $N_{cn}$ was observed from the marginal seas of China to the NWPO. For example, an $N_{cn}$ of $5.2 \pm 2.4 \times 10^3$ cm$^{-3}$ was recorded during the departure and return periods, including DOY 77–80 and DOY 109–112, respectively, against the $N_{cn}$ of $2.8 \pm 1.0 \times 10^3$ cm$^{-3}$ during the cruise over the NWPO on DOY 81-–108 (Fig. 2b), with the lowest $N_{cn}$ of $2.0 \pm 0.53 \times 10^3$ cm$^{-3}$ recorded during DOY 103–108. However, these values over the NWPO are approximately one order of magnitude larger than the values of <204 particle cm$^{-3}$ representing typical remote marine aerosols in the western Pacific measured by Mochida et al. (2011); <300 particle cm$^{-3}$ reported by Ueda et al. (2016), which was less affected by industrial activity in the tropical Pacific; and 248 particles cm$^{-3}$ derived from marine air masses over the northeast Pacific reported by Hudson and Xie (1999). Quinn et al. (2017) reviewed numerous observations of marine atmospheres and proposed that $N_{cn}$ beyond 540 particles cm$^{-3}$ is likely attributed to a substantial contribution from continental input. Long-term observation of the background marine aerosols in the Southern Ocean marine boundary layer (MBL) supports this assumption (Gras and Keywood, 2017). If the value of 540 cm$^{-3}$ represents the maximum contribution from marine natural sources, the mean $N_{cn}$ of $2.8 \times 10^3$ cm$^{-3}$ over the NWPO, during DOY 81–108, observed in this study indicates that at least 80% (on average) of $N_{cn}$ was contributed by continental inputs. Even for the period of DOY 103–108 with the smallest $N_{cn}$, the percentage of contribution from continental inputs using the same approach is still estimated to be as high as 73% on average, further implying the dominant contribution of continental input. Regarding the observed $N_{cn}$ alone, the remote marine atmosphere over the NWPO has been polluted to some extent in the spring; no traditionally defined clean marine atmosphere was present during that period.

The mean values of $N_{ccn}$ during the departure and return periods were $1.7 \pm 1.0 \times 10^3$ and $3.3 \pm 2.0 \times 10^3$ at SS levels of 0.2% and 0.4%, respectively; these SS levels represent moderate and high values in the remote marine atmosphere, respectively (Table S1). In contrast, the mean values of $N_{ccn}$ at these two SS levels decreased by approximately 60%, with values of $0.68 \pm 0.38 \times 10^3$ cm$^{-3}$ and $1.1 \pm 0.67 \times 10^3$ cm$^{-3}$, respectively, during DOY 81–108. Considering the mean values of $N_{ccn}$ and the availability of $N_{cn}$, $N_{ccn}$ in the atmosphere over the NWPO can be divided into four periods: DOY 81–86: Period 1, ending with no $N_{cn}$ data because of instrument malfunction; DOY 87–97: Period 2, ending with a large increase in $N_{ccn}$; DOY 98–102: Period 3, ending with a large decrease in $N_{ccn}$ to the low value; and DOY 103–108: Period 4. Lower values were recorded during Period 1, 2 and 4, and higher values were observed during Period 3 (Fig. 2a). For example, during Period 1, the mean $N_{ccn}$ was $0.56 \pm 0.12 \times 10^3$ cm$^{-3}$ at an SS of 0.2% and $0.84 \pm 0.20 \times 10^3$ cm$^{-3}$ at an SS of 0.4%. In contrast, during Period 3, several spikes in $N_{ccn}$ were observed, leading to higher values of $1.3 \pm 0.36 \times 10^3$ cm$^{-3}$ at an SS of 0.2% and $2.2 \pm 0.72 \times 10^3$ cm$^{-3}$ at an SS of 0.4% (Table 1). As shown in Fig. 1, the R/V traveled a long distance across the remote oceanic zones, approximately 1000 km or more from the continent during Periods 1, 2, and 4; however, it was close to Japan during Period 3, where the continental transport may have played a more important role in the number concentration.

When the values of $N_{ccn}$ at an SS of 0.2% in various marine atmospheres were used for comparison (Table 2), the lowest values were usually observed in the cleanest maritime atmospheres, e.g., the median daily $N_{ccn}$ were typically in the range of 15 to 30 cm$^{-3}$ in summer over the Arctic (Leck and Svensson, 2015). The mean $N_{ccn}$ increased slightly to approximately 70 cm$^{-3}$ during the summer over the Southern Ocean (Hudson et al., 1998). Even larger values were reported in atmospheres over the middle– and low–latitudes oceans, e.g., the median $N_{ccn}$ was 173 cm$^{-3}$ in June–July 2013 over the western North Atlantic (Kristensen et al., 2016), and the mean value of $N_{ccn}$ was approximately 100 cm$^{-3}$ in summer 2008 over the western Pacific (Mochida et al., 2011). In contrast to the observations made by Mochida et al. (2011), Roberts et al. (2006) reported that the $N_{ccn}$ influenced by aerosols of long-range transport from Asia was less than 300 cm$^{-3}$ in spring 2004 over the eastern Pacific Ocean. The larger $N_{ccn}$ measured over the NWPO in this study, at $0.68\pm0.38\times10^3$ cm$^{-3}$ at an SS of 0.2%, strongly indicates that the CCN was overwhelmingly derived from upwind continental input. Our larger $N_{ccn}$ is comparable to the observed values of $603\pm400$ cm$^{-3}$ in July and $660\pm624$ cm$^{-3}$ in August 2012 observed in the Bay of Bengal, India (Chate et al., 2017), but is smaller than that of approximately 1250 cm$^{-3}$ observed over the eastern Mediterranean from September to October 2007 (Bougiatioti et al., 2009).

A comparison of the values of $N_{ccn}$ at an SS of ~0.4% in various marine atmospheres (Table 2) revealed that this value observed over the NWPO in the present study is approximately one order of magnitude larger than the mean value of $N_{ccn}$ at an SS of 0.5% measured under the influence of the Asian continental air masses in the spring of 1994-1996 at Ogasawara Island (Nagao et al., 1999). This result implies a large increase in $N_{ccn}$ over the NWPO during the last decades. However, the larger $N_{ccn}$ over the NWPO is less than half that observed during the departure and return periods (i.e., $3.3\pm2.0\times10^3$ cm$^{-3}$ with an SS of 0.4%) and is comparable to the observed $N_{ccn}$ in the spring of 2005 at Jeju Island, Korea (Kuwata et al., 2008) and the values recorded in the spring of 2001 at the lower altitude over the East China Sea (Adhikari et al., 2005). The higher $N_{ccn}$ values recorded during the departure/return period and at Jeju island (Kuwata et al., 2008) imply the possibility of large removal of atmospheric particles, which may enable activation as CCN during long-range transport (Guo et al., 2016). It is interesting to note that the value of $1.1\pm0.67\times10^3$ cm$^{-3}$ $N_{ccn}$ over the NWPO in this study is also comparable to those recorded at the same SS level in the Bay of Bengal (Chate et al., 2017), but is smaller than that of $2340\pm480$ cm$^{-3}$ at an SS of 0.38% observed in the September 2011 and 2012 in the remote marine atmosphere over the South China Sea under the condition of smoke intrusion of (Atwood et al., 2017).

According to the results of 72-h air mass backward trajectory analysis over the NWPO at 1000 m (Fig. S4), the air masses originated mostly from either Siberia or Northeast China and then passed over Japan to the reception zones. These results imply that the large increases in $N_{cn}$ and $N_{ccn}$ over the NWPO may have been influenced by the Asian continental outflow of air pollutants. In the remaining cases, air masses at 1000 m originated from oceanic zones. However, owing to the strong westerly wind in the upper free troposphere (FT), the long-range transport of air pollutants from Asian continents aloft may have been mixed downward to contribute largely to the increased $N_{cn}$ and $N_{ccn}$ therein.

## 3.2 Spatiotemporal variation of CCN activity and factor analysis

The AR was calculated as $N_{ccn}$ at a certain SS level divided by $N_{cn}$ during the entire cruise period during DOY 81-108, 2014, over the NWPO (Fig. 2c). The values of 0.30 $\pm$0.11 at an SS of 0.2% and 0.46 $\pm$0.19 at an SS of 0.4% in this study are almost the same as those reported previously, such as 0.28 $\pm$0.17 at SS = 0.2% and 0.43 $\pm$0.24 at SS = 0.4% obtained in the spring of 2013 at an upwind semi-urban site in Qingdao, China (Li et al., 2015) as well as those obtained in other semi-urban atmospheres (Rose et al., 2010; Leng et al., 2014). However, the values of AR at an SS of 0.6% were only ~0.3 in the spring 2004–2010 in Seoul; the same value of AR at an SS of 0.3% was recorded in the spring of 2001 near the Sea of Japan (Adhikari et al., 2010; Kim et al., 2014). In terms of the four periods, the smaller values of AR (0.21 $\pm$0.06 at an SS of 0.2% and 0.32 $\pm$ 0.09 at an SS of 0.4%) were observed during Period 1, which is similar to that recorded in Period 2 and 4 (Table 1). The larger values, 0.38 $\pm$0.11 at an SS of 0.2% and 0.64 $\pm$0.18 at an SS of 0.4%, occurred during Period 3, which is close to those of 0.43 $\pm$0.13 at an SS of 0.2% and 0.61 $\pm$0.15 at an SS of 0.4% recorded during a moderately heavy pollution event in the spring of 2013 in Qingdao (Li et al., 2015). These results imply that the aerosol particles during Period 3 were aged to a high extent. This result will be discussed subsequently.

As proposed in previous studies, e.g., Dusek et al. (2006) and Kalivitis et al. (2015), the total number concentration ($N_{>Dp}$) of particles larger than a threshold diameter ($D_p$) can be used as a proxy for $N_{ccn}$. Specifically, aerosol particles with sizes exceeding 60–70 nm could be activated as CCN at an SS of 0.4% (Dusek et al., 2006). In this study, $N_{>Dp}$ with $D_p$ varying from 50 nm to 80 nm was calculated and a linear correlation was conducted with the values of $N_{ccn}$ measured at an SS of 0.4%. Good correlation was obtained between $N_{ccn}$ and $N_{>60\,nm}$, with a slope of 0.98 closer to unity and $R^2$=0.94 (Fig. 3). It should be note that the data associated with BB and dust aerosols as well as those suspected to be BB or dust aerosols, were excluded from the analysis. The corresponding effective hygroscopicity parameter ($\kappa$) was further estimated to be 0.40, which is close to the $\kappa$-value of continental atmospheric aerosols (~0.3) and smaller than that of marine atmospheric aerosols (~0.7) (Kreidenweis et al., 2009; Pöschl et al., 2009.; Rose et al., 2010). This result was expected because of the overwhelming continental contribution to $N_{cn}$ measured over the NWPO in this study. Mochida et al. (2010a) measured the $\kappa$-value as about 0.50 $\pm$0.05 at Okinawa Island, Japan, on 3–12 April 2008, which implies a large contribution from ammonium sulfate aerosols. Iwamoto et al. (2016), however, reported a lower $\kappa$-value around 0.27 $\pm$ 0.21 at the tip of Noto Peninsula, Japan, on 3–29 October 2012, in association with a high proportion of organics in the atmospheric particles.

The same test was conducted to establish a correlation between $N_{ccn}$ at an SS of 0.2% and $N_{>Dp}$. We obtained the best correlation between $N_{ccn}$ and $N_{>92.5\,nm}$, with $R^2$=0.92 and a slope of 1.40 (Fig. S5). The FMPS has a limitation of a low resolution in the size range of >100 nm; no exact values of $D_p$ and $N_{>Dp}$ corresponding to $N_{ccn}$ at an SS of 0.2% were further calculated.

The ratios of $N_{>60\,nm}/N_{ccn}$ were separately considered during the four periods, i.e., 1.06 $\pm$0.43 and 1.04 $\pm$0.24 at SS = 0.4% during Periods 3 and 4, respectively (Fig. 4). Ammonium sulfate had the critical diameter ($D_c$) = 53 nm at SS=0.4% and was likely a major contributors of atmospheric particles larger than 60 nm during Periods 3 and 4. The standard deviation

reflected its variation in relative contribution to some extent. The values of $N_{>60 nm}/N_{ccn}$ increased to $1.30 \pm 0.36$ and $1.64 \pm 0.62$ at SS = 0.4% during Periods 1 and 2, respectively (Fig. 4). A few values of $N_{>60 nm}/N_{ccn}$ beyond 1.5 corresponded to smaller values of $N_{ccn}$, i.e., $<2.5 \times 10^3$ cm$^{-3}$, implying that the aerosol particles were less hygroscopic. Parts of the aerosols were from BB or dust aerosols, as will be discussed subsequently.

### 3.3 Particle number size distributions and spatiotemporal variations in particles of different sizes

Examination of the particle number size distributions (Fig. 1 and Fig. S6) revealed a trough present in the range of 50–90 nm during the entire campaign. This trough is conventionally referred to as the Hoppel effect associated with atmospheric particles modified by nonprecipitating clouds (Hoppel et al., 1986, 1994a, 1994b). The in-cloud processing could result more aged and hygroscopic atmospheric particles. The accumulation mode of atmospheric particles with diameters larger than 90 nm ($N_{>90 nm}$) showed a clear decrease from $4.0 \pm 1.4 \times 10^3$ cm$^{-3}$ on DOY 78 to $1.1 \pm 0.4 \times 10^3$ cm$^{-3}$ on DOY 80 when the cruise was relatively far from the continent, then oscillated around the lower value during Periods 1 and 2. The $N_{>90 nm}$ evidently increased to $2.3 \pm 0.6 \times 10^3$ cm$^{-3}$ during Period 3 and then decreased to $1.1 \pm 1.1 \times 10^3$ cm$^{-3}$ during Period 4. During the return trip, when approaching to the continent, the $N_{>90 nm}$ increased up $2.2 \pm 0.4 \times 10^3$ cm$^{-3}$ on DOY 110 and to $4.7 \pm 1.5 \times 10^3$ cm$^{-3}$ on DOY 111 with the increase in continental input.

The nucleation–mode particle concentrations in the size range <30 nm ($N_{<30 nm}$) generally dominated over the $N_{>90 nm}$ in the atmosphere over the NWPO except during the departure and return periods and Period 3. Unlike the $N_{>90 nm}$, the $N_{<30 nm}$ did not show a clear decrease from the marginal seas to the NWPO, i.e., the values of $N_{<30 nm}$ were $1.8 \pm 0.7 \times 10^3$ cm$^{-3}$ on DOY 78 and $1.9 \pm 0.3 \times 10^3$ cm$^{-3}$ on DOY 80. The $N_{<30 nm}$ narrowly oscillated between $1.5 \pm 0.2 \times 10^3$ cm$^{-3}$ during Period 1 and $1.4 \pm 0.1 \times 10^3$ cm$^{-3}$ during Period 2 and then decreased to $1.1 \pm 0.3 \times 10^3$ cm$^{-3}$ during Period 3; however, the corresponding $N_{>90 nm}$ showed an evident increase, as mentioned previously. The $N_{<30 nm}$ was $1.0 \pm 1.0 \times 10^3$ cm$^{-3}$ during Period 4. Strong continental input led to a sharp increase in the $N_{<30 nm}$ from $1.0 \pm 0.2 \times 10^3$ cm$^{-3}$ on DOY 110 to $3.7 \pm 1.5 \times 10^3$ cm$^{-3}$ on DOY 111. It should be noted that the data containing the ship's self-emissions were exhaustively excluded; thus, we hypothesize that the small decrease in the $N_{<30 nm}$ over the NWPO might have been caused by dynamic sources forming the nucleation mode particle pool along the track.

The Aitken mode, in the size range of 30–60 nm, was clearly identified during the departure and return periods and in Period 3, but it largely decreased and sometimes was undetectable during Periods 1, 2 and 4. No significant change was detected in number concentrations of particles with diameters between 30 nm and 60 nm ($N_{30-60 nm}$) from the marginal seas to the open ocean during DOY 78–80; that is, the $N_{30-60 nm}$ was $1.9 \pm 0.6 \times 10^3$ cm$^{-3}$ on DOY 78 and $1.7 \pm 0.2 \times 10^3$ cm$^{-3}$ on DOY 80, which is similar to the $N_{<30 nm}$. However, the $N_{30-60 nm}$ then sharply decreased approximately 40%, down to $1.1 \pm 0.2 \times 10^3$ cm$^{-3}$ during Period 1 and $1.0 \pm 0.1 \times 10^3$ cm$^{-3}$ during Period 2. The larger decrease in the $N_{30-60 nm}$ relative to the $N_{<30 nm}$ during the two periods can be attributed to either fewer nucleation–mode particles being able to grow in the Aitken–mode particles or to more Aitken–mode particles being removed by the Hoppel effect. The $N_{30-60 nm}$ of $1.0 \pm 0.4 \times 10^3$ cm$^{-3}$ during Period 3

was almost the same as that during Periods 1 and 2, although the latter varied widely. The $N_{30-60 \text{ nm}}$ further decreased to $0.7 \pm 0.7 \times 10^3$ cm$^{-3}$ during Period 4 but varied more widely until the cruise returned and was close to the continent, showing a value of $2.7 \pm 0.8 \times 10^3$ cm$^{-3}$ on DOY 112. The ratios of $N_{30-60 \text{ nm}}/N_{<30 \text{ nm}}$ were close to unity during DOY 78–80 and decreased afterward, ranging from 0.6 to 0.8 during each period. This may be explained by the two aforementioned possibilities.

As reported by Vu et al. (2015), the particle number size distributions in the marine atmospheric boundary layer usually show two modes, Aitken mode and accumulation mode, with a nucleation mode observed occasionally (Koponen et al., 2002; Ueda et al., 2016; Zhu et al., 2019). For example, the particle size number concentrations exhibited a bimodal distribution with an Aitken mode (~ 50 nm) and an accumulation mode (150–180 nm) during the fall campaign over the western North Pacific in 2008 (Mochida et al., 2011). Bimodal distributions were also reported during a winter campaign over the tropical and

subtropical Pacific Oceans from 2011 to 2012 (Ueda et al., 2016) and during a campaign over the western North Atlantic in June–July 2013 (Krstensen et al., 2016). However, the Aitken mode and the accumulation mode were sometimes overlapped in the particle number size spectra measured over marginal seas influenced by polluted air masses (Lin et al., 2007; Nair et al., 2013; Zhu et al., 2019).

The overall particle size distribution was dominated by the nucleation mode during Periods 1, 2, and 4 but by the

accumulation mode during Period 3 (Fig. 1). Considering the more important roles of accumulation–mode particles, compared with those of nucleation mode in modulating the CCN, the mean value of the associated AR during Period 3 therefore was twice that during Period 1 even though the mean value of $N_{cn}$ during Period 3 was only 25% higher than that of Period 1. Most of Aitken–mode particles were likely unable to be activated as CCN at an SS $\leq$ 0.4% (Dusek et al., 2006). Recently, Fan et al. (2018) reported that atmospheric particles smaller than 50 nm can be activated to form additional cloud droplets in a low-

aerosol environment, where deep convective clouds can develop very high vapor supersaturation. In the NWPO, very high vapor supersaturation could also occur and might cause the Aitken mode particles to be activated into CCN. This hypothesis, however, requires more evidence for confirmation.

### 3.4 New particle formation events in the NWPO

During the entire campaign over the NWPO, only two new particle formation (NPF) events were observed on DOY 98

and DOY 103. Unlike previous studies in the marginal seas of China (Liu et al., 2014; Meng et al., 2015) and in the south subtropical Pacific (Ueda et al., 2016), the events of this study lasted for a short time. The newly formed particles apparently grew to a size smaller than 20 nm during the events. The first NPF event was observed from 05:43 to 06:00 (UTC+8) on DOY 98 with the ~8 nm minimum diameter of particles growing to ~14 nm (Fig. 5a). The averaged value of $N_{<30 \text{ nm}}$ during the NPF event nearly doubled the value observed 1 h before or after the NPF event, although the average value of $N_{30-100 \text{ nm}}$ during the

NPF event was almost unchanged (Fig. 5b). From 05:00 to 08:00 (UTC+8), the wind direction was almost constant, and the wind speed narrowly varied (Fig. 5c). In terms of the second new particle event, the signal of newly formed particles was intermittently observed for approximately 1 h on DOY 103 (Fig. 5d).

In the literature, NPF events in remote marine atmospheres usually occur aloft rather than in the lower layers of the MBL. Wiedensohler et al. (1996) and Buzorius et al. (2004) reported that ultrafine particles are produced above or in the upper layers of the MBL and are mixed downward to the atmosphere above sea level. Clark et al. (1998) and McNaughton et al. (2004) also reported a nucleation event above the MBL during aircraft observations over the Pacific Ocean. Recent measurements further support that NPF events most likely occur in the FT over different oceanic zones (Dadashazar et al., 2018; Rose et al., 2015; Sanchez et al., 2018; Takegawa et al., 2014). Several factors such as the lower temperatures and lower relatively humility, lower condensation sinks, and mixed precursors from the continental and marine sources lower in the FT have also been offered as explanations for the NPF occurrence therein. Regarding the pool of nucleation–mode particles in the atmosphere over the NWPO mentioned previously and considering that only two NPF events were observed during dozens of minutes, we can reasonably infer that NPF events very likely occur frequently above or in the upper layers of the MBL. When the NPF occurs aloft, two scenarios could lead to the unique observations over the NWPO. In limited cases, where newly formed particles require a short time to be mixed downward, the NPF events can then be detected in the lower layer of MBL. In the other scenario, the newly formed particles require a long time to be mixed downward because they were originally formed above the MBL and survived from coagulation and scavenging. In this scenario, no NPF event can be detected in the lower layer of the MBL with a time delay relative to that above the MBL. However, the pre–existing new particles might lead to much larger $N_{<30\ nm}$ in the atmosphere than that observed in the marine natural background with narrow variation over a large spatial scale. Based on the calculated air mass back trajectories over the NWPO at 1000 m above the sea surface (not shown), air masses from above the MBL and even those from the FT can be mixed down to the lower layer of the MBL. However, the duration time varies greatly among cases.

Regarding the increase in $N_{cn}$ by NPF events, a few studies have proposed that nucleation–mode particles can increase in size, even reaching the CCN size in the FT (Rose et al., 2015; Sanchez et al., 2018), with the growth occurring during the subsidence process from FT to the MBL. Sanchez et al. (2018) estimated that the contributions of NPF in the FT to the $N_{ccn}$ at an SS of 0.1% in the clean marine atmosphere over the North Atlantic are 31% and 33% in late autumn and late spring, respectively. Merikanto et al. (2009) reported that 55% of CCN at an SS of 0.2% in the MBL are from nucleation, with 45% entrained from the FT and the reminding 10% nucleated directly in the boundary layer. However, growth of newly formed particles to the CCN size was not observed in this study.

### 3.5 The influence of wind speed on $N_{ccn}$

As discussed in section 3.1, the major source of $N_{ccn}$ is continental input. In addition, wind speed may modulate the contribution of sea salt and vertical turbulence transport to the $N_{ccn}$. According to previous studies, the logarithmic relationship between wind speed and film–dropped sea salt is log $N_{film} = 0.095\ U_{10} + 0.283$, where $N_{film}$ is the number concentration (cm$^{-3}$) and $U_{10}$ is the wind speed at 10 m above the sea surface (O'Dowd et al., 1993, 1997; de Leeuw et al., 2011). It is generally accepted that a particle size larger than 0.05 μm can be activated as CCN in the marine atmospheric environment, where SS of 0.2–0.3% is regularly reached. Under high wind speeds, sea salt aerosols could be important contributors to CCN in the MBL.

For example, O'Dowd et al. (1997) reported that the sea salt CCN concentration increases with an increase in wind speed, reaching 150 cm$^{-3}$ at an SS of 0.3% and a wind speed of 20 m/s.

To study the relationship between $N_{ccn}$ and wind speed, we plotted a time series of the $N_{ccn}$ at an SS of 0.2% to determine the wind speed and wind direction at ~10 m above the sea surface in the NWPO during the four periods of the cruise (Fig. 6). During Period 1, the wind speed varied from 0.9 m/s to 11.6 m/s with a mean value of $6.7 \pm 2.8$ m/s (Fig. 6a). The sea salt CCN concentration at an SS of 0.2% was reported as ~10 cm$^{-3}$ at the wind speed of 10 m/s (O'Dowd et al., 1997), thereby accounting for less than 2% of $N_{ccn}$ ($0.56 \pm 0.12 \times 10^3$ cm$^{-3}$, as discussed in section 3.1). The wind speed varied in a broad range of 0.8 m/s to 18.3 m/s with a mean value of $8.3 \pm 3.7$ m/s during Period 3 (Fig. 6c). If the value of 100 cm$^{-3}$ at a wind speed of 20 m/s at an SS of 0.2% represents the maximum contribution from sea salt CCN (O'Dowd et al., 1997) for Period 3 with the highest $N_{ccn}$ of $1.3 \pm 0.36 \times 10^3$ cm$^{-3}$, as discussed in section 3.1, the contribution of sea salt CCN accounted for less than 8%. These results imply a minor contribution to $N_{ccn}$ from sea salt.

Regarding of the vertical distribution of the $N_{ccn}$ over the marine atmosphere, three scenarios are hypothesized: 1) the $N_{ccn}$ aloft was larger than that in the atmosphere near sea level; 2) the $N_{ccn}$ in the vertical direction was homogenous; 3) or the $N_{ccn}$ aloft was lower than that in the atmosphere near sea level. Varying wind speeds may change the convection, which in turn affects $N_{ccn}$ in the atmosphere near sea level. For example, Clarke et al. (2013) reported that CCN activated in MBL clouds is strongly influenced by entrainment from the FT. Zheng et al. (20118) also argued that entrainment of FT aerosols is a vital source of accumulation mode particles over the eastern North Atlantic, which could be easily activated as CCN. The common high-pressure system over the NWPO may promote the subsidence of CCN from the FT. In this study, $N_{ccn}$ had no significant correlation with wind speed at the 95% confidence level when all data were analyzed. Our results are consistent with those of previous studies: Ueda et al. (2016) reported no positive correlation between the number concentrations of particles <500 nm and wind speeds in the tropical and subtropical Pacific Ocean. However, an obvious correlation between $N_{ccn}$ and wind speed was observed during a few short periods. For example, a positive correlation was obtained during DOY 102–103 (Fig. 6c). This implies that the enhanced vertical transport of high $N_{ccn}$ aloft is likely attributed to increased wind speed because a high wind speed generally dilutes the air mass to some extent (Hudson and Xie, 1999). On the contrary, a negative correlation existed during DOY 92-94 (Fig. 6b). This phenomenon was attributed low $N_{ccn}$ aloft, possibly even lower than at the sea surface (Hudson et al., 1998; Hudson and Xie, 1999; Kim et al., 2014); therefore, the enhanced turbulence might cause cleaner air to be mixed downward, resulting in a subsequent decrease in $N_{ccn}$. We also found that $N_{ccn}$ had no evident change with a large changed in wind speed during a few periods, e.g., DOY 99–100 (Fig. 6c), which would imply an even distribution in the vertical direction (Hudson et al., 1998). According to the vertical backward air mass trajectories (Fig. S7), the air masses are transported mostly from the Asian continent at high altitudes (>2000 m a.m.s.l.) to the reception zones, indicating that air masses are affected by the entrainment of FT aerosols. Therefore, it is reasonable to argue that the $N_{ccn}$ mixed downward from the FT may be an important source of $N_{ccn}$ in the MBL over the NWPO. However, modeling studies are needed in the future to quantify the contribution.

## 3.6 Evidence and contributions of CN and CCN from BB and dust aerosols

The long-range transport of BB and dust aerosols from the Asian continent to the NWPO has attracted significant attention (Huebert, et al., 2003; Luo et al., 2016; Fu et al., 2018). For example, inorganic ions in TSP samples collected by Fu et al. (2018) on the same cruise as this study indicate that the remote NWPO could be influenced by BB aerosols. In addition, Luo et al. (2016), also on the same cruise, reported the occurrence of dust and dust mixed with anthropogenic pollutants over the NWPO. Thus, the contributions of the two sources to $N_{cn}$ and $N_{ccn}$ are examined below.

According to the three–day back trajectories at 1000 m, most of the air masses originated from either Siberia or Northeast China during this cruise campaign (Fig. S4). In the spring of 2014, satellite data showed a large number of fire events occurring on these upwind continents (Fig. S8). A TSP sample was collected during the period from 11:10 on DOY 101 to 23:30 on DOY 102 (UTC+8), in which the mass concentration of levoglucosan (LEVO), a BB tracer, was as high as 65 ng m$^{-3}$. This value is one order of magnitude higher than that recorded over the islands of Chichi-jima in the spring 2001 to 2004 and Okinawa in 2009–2012, at 1.0 ng m$^{-3}$ and 3.09 $\pm 3.70$ ng m$^{-3}$, respectively (Mochida et al., 2010b; Zhu et al., 2015). The time series of $N_{cn}$, bulk $N_{ccn}$, $N_{ccn}/N_{cn}$, $N_{>60 nm}/N_{ccn}$, and geometric median diameter of 100–334 nm particles (GMD$_{100-334}$) at an SS of 0.4% during Period 3 (Fig. 7) indicate that the BB event occurred exactly during the period from 22:19 on DOY 101 to 03:49 on DOY 102 (UTC+8), belonging to Period 3 (DOY 98–102) defined in this study. The averaged value of $N_{cn}$ during these 5.5 h (green shading in Fig. 7a) nearly doubled the mean value during the entire Period 3. However, the averaged value of $N_{ccn}$ was slightly smaller than the mean value of Period 3 (green shading in Fig. 7a). At SS =0.4%, the averaged value of $N_{ccn}/N_{cn}$ decreased 38% against the average value during the entire Period 3, whereas the averaged value of $N_{>60 nm}/N_{ccn}$ was approximately 80% higher than that of Period 3 (green shading in Fig. 7b). This suggests that a large portion of BB aerosol particles were not activated as CCN. Owing to the limited size cut of FMPS for submicron particles, only the range of the κ value can be roughly estimated with values <0.1. The κ values from the different BB aerosols have been reported as 0.2 $\pm 0.1$ after a few hours of photochemical processing (Engelhart et al., 2012). Assuming that $N_{cn}$ and $N_{ccn}$ for these 5.5 h (i.e., 22:19 on DOY 101 to 03:49 on DOY 102) were composed completely of BB aerosols, the $N_{cn}$ and $N_{ccn}$ during that period accounted for 18% and 9%, respectively, relative to the corresponding total recorded on DOY 101–102. We also found that the GMD$_{100-334}$ of the particles during the BB aerosol event reached the minimum value, which was decreased by ~10 nm relative to that recorded during the entire Period 3 (green shading in Fig. 7c). In addition to this sample collection from 22:19 on DOY 101 to 03:49 on DOY 102, the LEVO concentrations in the other TSP samples were less than 13.7 ng m$^{-3}$ with an average of 4.5 ng m$^{-3}$, indicating a large decrease in loadings of BB aerosols. This suggests relatively smaller contributions of BB aerosols to $N_{cn}$ and $N_{ccn}$ during the other sampling periods.

During the period from 01:44 to 07:38 DOY 98 (UTC+8), the NASA LiDAR (Fig. S9) detected a signal of dust sweeping over the measurement zone. The averaged value of $N_{cn}$ during the dust event was nearly twice that of Period 3, with the averaged value of $N_{ccn}$ decreasing by 27% (gray shading in Fig. 7a). Lee et al. (2009) modelled the influence of dust events on CCN and reported a maximum decrease of 20%. The averaged value of $N_{>60 nm}/N_{ccn}$ was computed as 2.3 and was even larger

than that of 1.8 for BB aerosols, and the estimated $\kappa$ value of dust aerosols was smaller than 0.1. Although the uptake of sulfuric and nitric acids on dust aerosols may increase their hygroscopicity (Manktelow et al., 2010; Matsuki et al., 2010), they apparently had a negligible influence on the contribution to $N_{ccn}$ at SS = 0.4%. Additionally, assuming that $N_{cn}$ and $N_{ccn}$ from 01:44 to 07:38 on DOY 98 (UTC+8) were contributed completely by dust aerosols, the $N_{cn}$ and $N_{ccn}$ during these few hours accounted for 58% and 34%, respectively, relative to the corresponding totals of $N_{cn}$ and $N_{ccn}$ on DOY 98. In terms of the entire cruise period, $N_{>60\,nm}/N_{ccn}$ beyond 1.5 clearly deviated from the general trend (Fig. 3). Thus, if $N_{>60\,nm}/N_{ccn}$ was greater than 1.5, which is smaller than the value of 2.3 for $N_{>60\,nm}/N_{ccn}$ during the dust event and 1.8 for $N_{>60\,nm}/N_{ccn}$ during the BB event, it was considered to have an impact from dust and BB aerosols or suspected dust and BB aerosols, and the subsequent summation of $N_{ccn}$ and $N_{cn}$ during these days over the NWPO accounted for less than 10% of the total $N_{ccn}$ and $N_{cn}$ during this period. This indicates a relatively minor contribution of dust and BB aerosols to $N_{cn}$ and $N_{ccn}$ on a monthly scale.

## 4 Conclusions

A field campaign was conducted to study the influence of the outflow from the Asian continent on $N_{cn}$, $N_{ccn}$, and the number size distribution of aerosol particles over the NWPO during DOY 77–112. The average $N_{cn}$ and $N_{ccn}$ were approximately one order of magnitude higher than those observations previously reported in the remote marine atmosphere during other seasons or in the atmosphere of other remote marine areas, implying that the NWPO received more continental input owing to the rapid increase in emissions of air pollutants in recent years during spring. The CCN activities were almost the same as those of the atmosphere of semi-urban areas, which implies that the aerosol particles over the NWPO were aged to some extent. The $N_{>60\,nm}$ had good correlation with $N_{ccn}$ at an SS of 0.4%, with an $N_{>60\,nm}/N_{ccn}$ ratio close to 1, at 0.98. When we used $D_p = 60$ nm as a proxy for the $D_c$, the $\kappa$ was estimated to be 0.40. The value slightly increased against the average value of 0.3 widely observed in upwind continental atmospheres, implying that additional aerosol aging during the long-range transport in the marine atmosphere had a minor influence on $k$.

When the particle number size distribution was examined, a bimodal size distribution pattern was generally observed, and the particle size distribution was dominated by the nucleation mode in Period 1 and the accumulation mode in Period 3. The $N_{<30\,nm}$ narrowly varied from the marginal sea to the NWPO, whereas the $N_{>90\,nm}$ changed vastly among periods. This implies that relatively stable sources generated the smaller size particles along the track. We found that approximately one–third of $N_{cn}$ was contributed from $N_{<30\,nm}$. Two NPF events were measured during the observation period. The vertical transport of newly formed particles to the lower layer of the MBL appeared to be important sources of nucleation mode particles measured along the track.

When the data measured during the entire campaign were used, no positive correlation between $N_{ccn}$ and wind speed was found owing to the minor contribution of the wind-driven particle production at the ocean surface. However, good positive or negative correlations between $N_{ccn}$ and wind speed were obtained in a few short periods. These correlations can be attributed likely to the possibly large gradient of $N_{ccn}$ in the vertical direction together with the enhanced or reduced vertical transport.

Intrusion events of BB and dust aerosols were occasionally observed over the NWPO. The $N_{ccn}$ slightly decreased with an increase in $N_{cn}$ during dust and BB events. Considering the lower $N_{ccn}/N_{cn}$ ratios and the higher $N_{>60\ nm}/N_{ccn}$ during those events, we speculate that a large proportion of particles can not to be activated as CCN. Moreover, we detected a minor contribution of dust and BB aerosols to CN and CCN in the number concentration on a monthly time scale.

**Data availability.**   The data of this paper are available upon request (contact: Juntao Wang, wangjuntao@stu.ouc.edu.cn).

**Competing interests.**   The authors declare that they have no conflict of interest.

**Acknowledgements**

We would like to thank the support from the National Key Research and Development Program in China (No.2016YFC0200504) and the Natural Science Foundation of China (Grant No. 41576118).

**Authors contribution**

Juntao Wang, Yang Gao and Xiaohong Yao designed the project, processed the data and performed the analysis. Yanjie Shen, Kai Li, and Huiwang Gao were involved in planning and supervised the manuscript. Juntao Wang prepared the
manuscript with contributions from all co-authors.

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

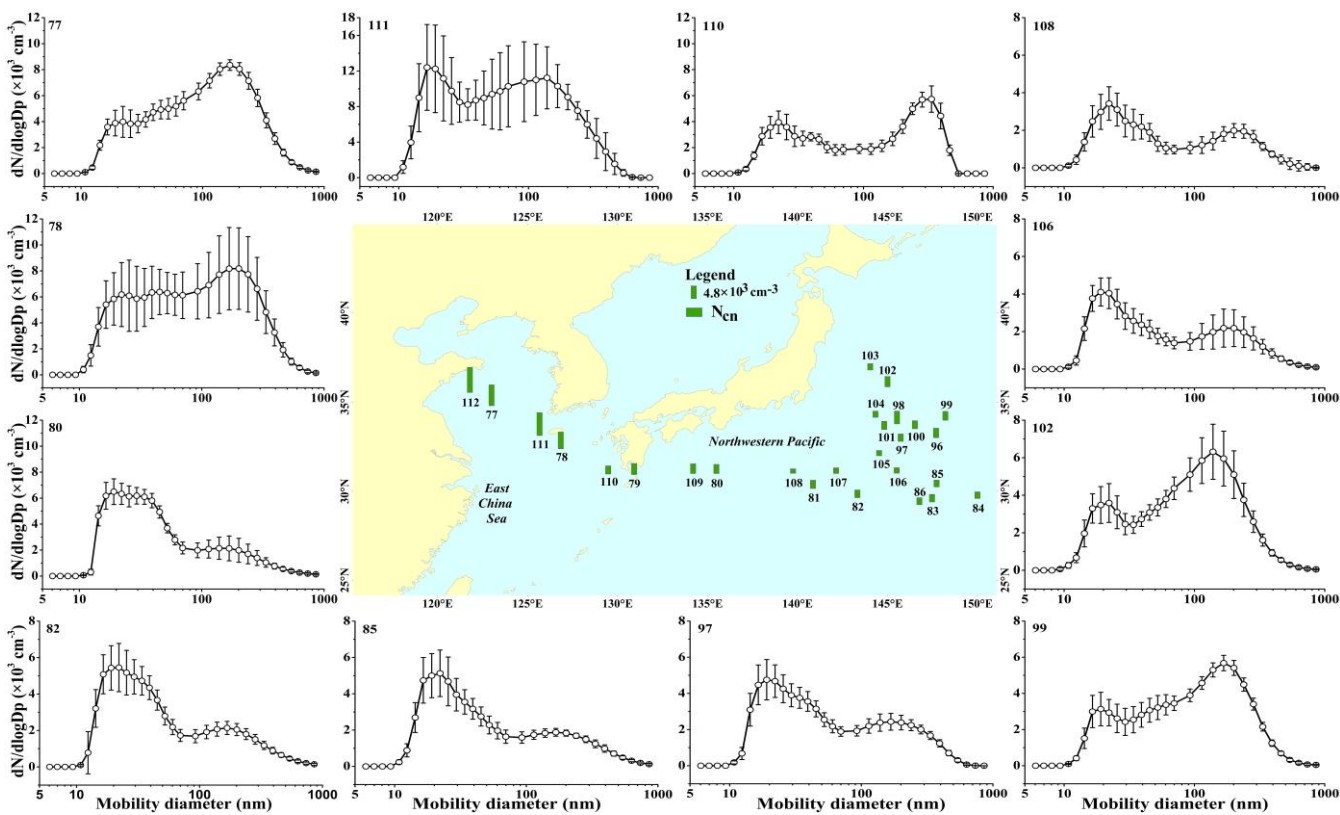

**Figure 1** Geographical distributions of daily $N_{cn}$ and number-size distribution of aerosol particles measured over the NWPO during DOY 77-112, 2014. The entire cruise was divided into five periods, including departure and return periods (77-80,109-112), Period 1(81-86), Period 2 (87-97), Period 3 (98-102) and Period 4 (103-108). For each period, the mode distribution is relatively comparable; thus, only a few days were selected to show the number size distributions, with the cruise day showing on the top left, corresponding to the number in the middle map.

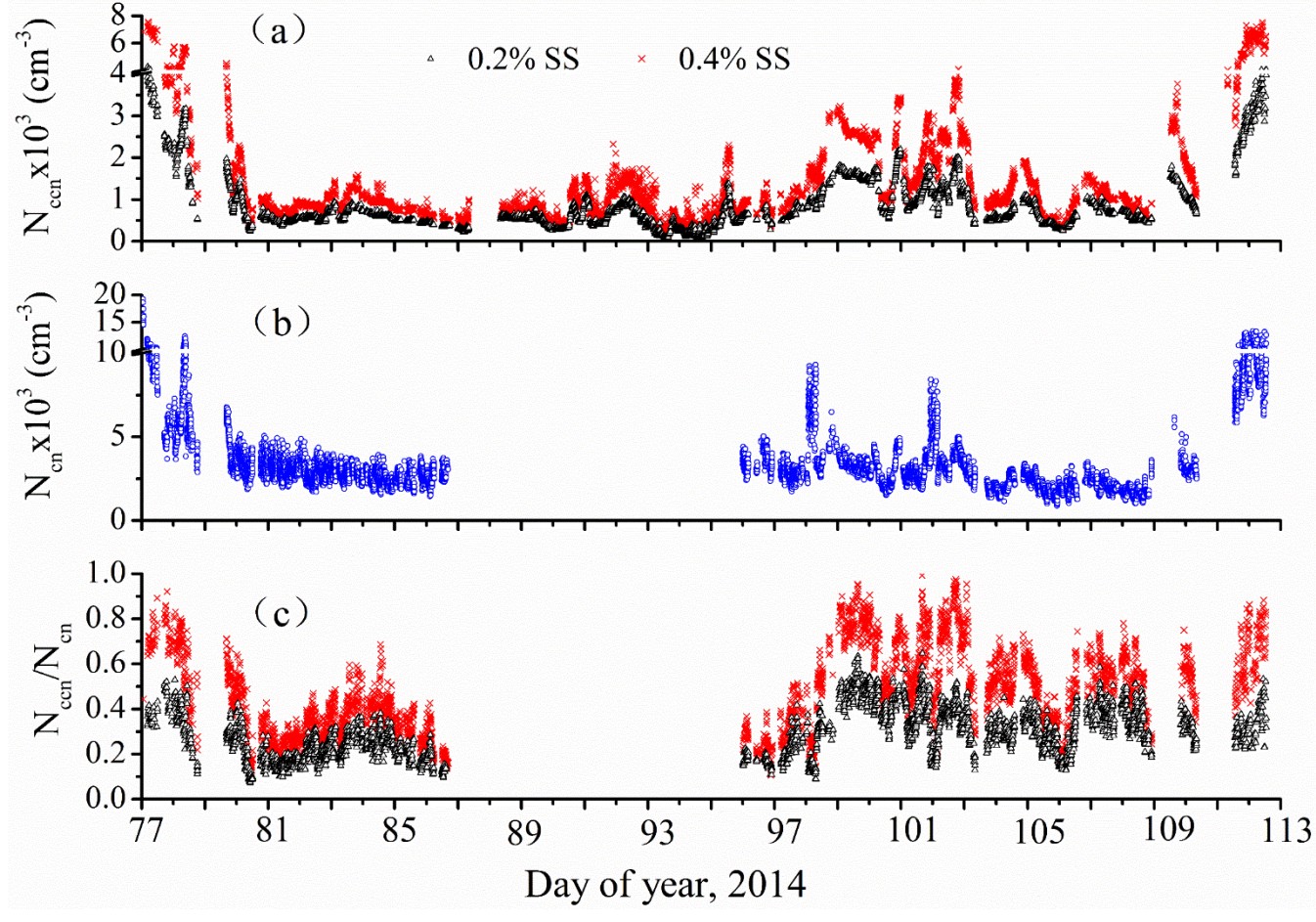

**Figure 2** Time series of minutely averaged $N_{cn}$, bulk $N_{ccn}$ and $N_{ccn}/N_{cn}$ at SS of 0.2% and 0.4% during DOY 77-112, 2014.

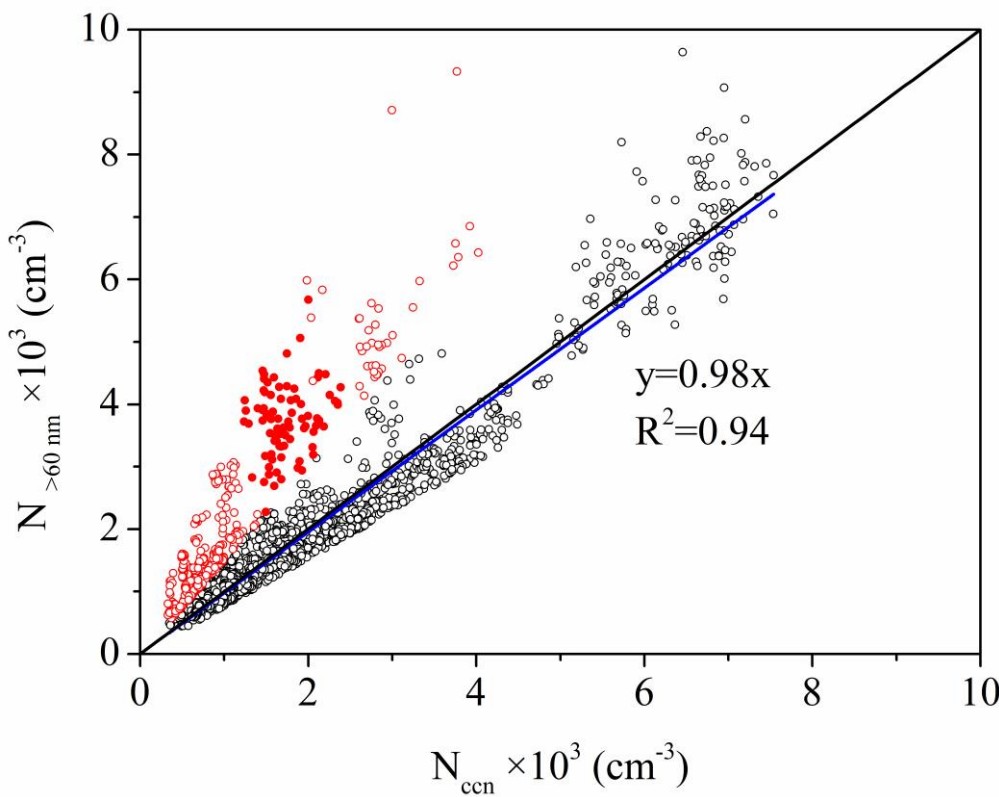

**Figure 3** Scatterplots of $N_{ccn}$ at SS of 0.4% versus $N_{>60\ nm}$. (BB and dust aerosols are shown in full red cycles with empty red cycles representing suspected either BB or dust aerosols; the black line represents the 1:1 relationship, and the blue line shows the best fit using the data shown as black empty cycles.)

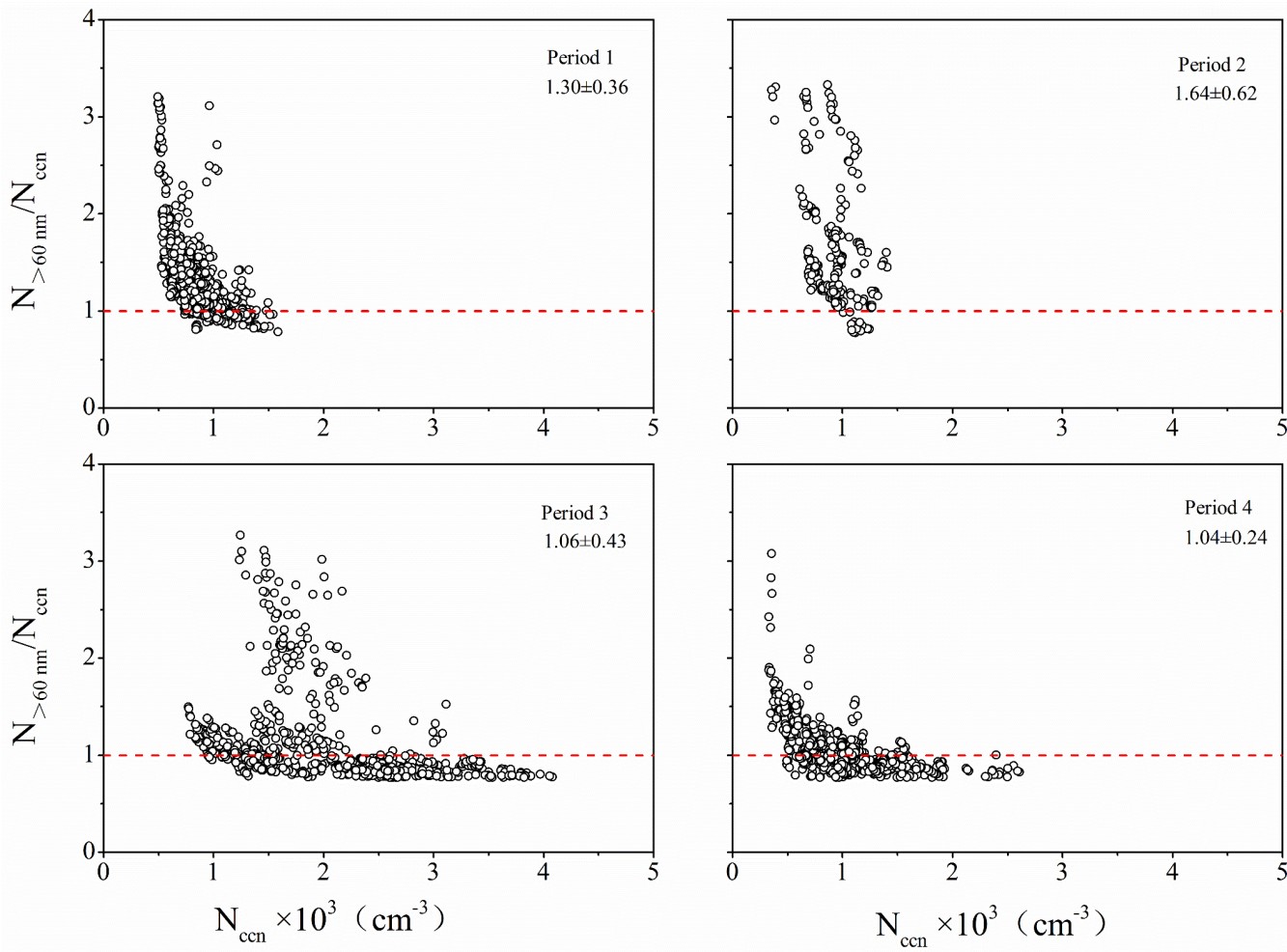

**Figure 4** The ratios of $N_{>60\,nm}/N_{ccn}$ as a function of $N_{ccn}$ at SS of 0.4% in different periods.

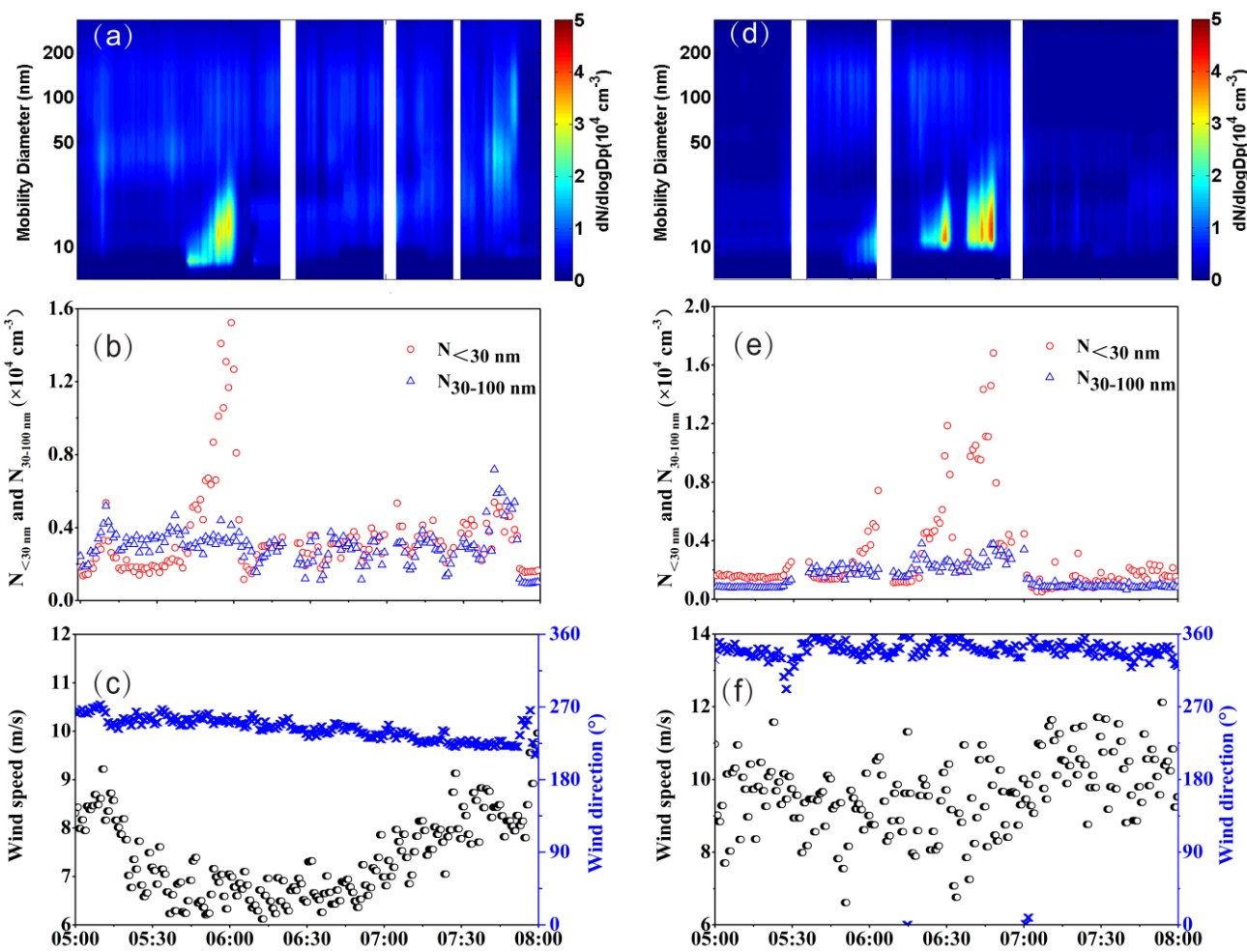

**Figure 5** Temporal variations of particle number size concentrations, particle concentrations, and meteorological parameters for NPF events observed on DOY 98 (a, b, c) and DOY 103 (d, e, f) 2014. Data points suffering from ship self-emissions were removed to avoid clustering.

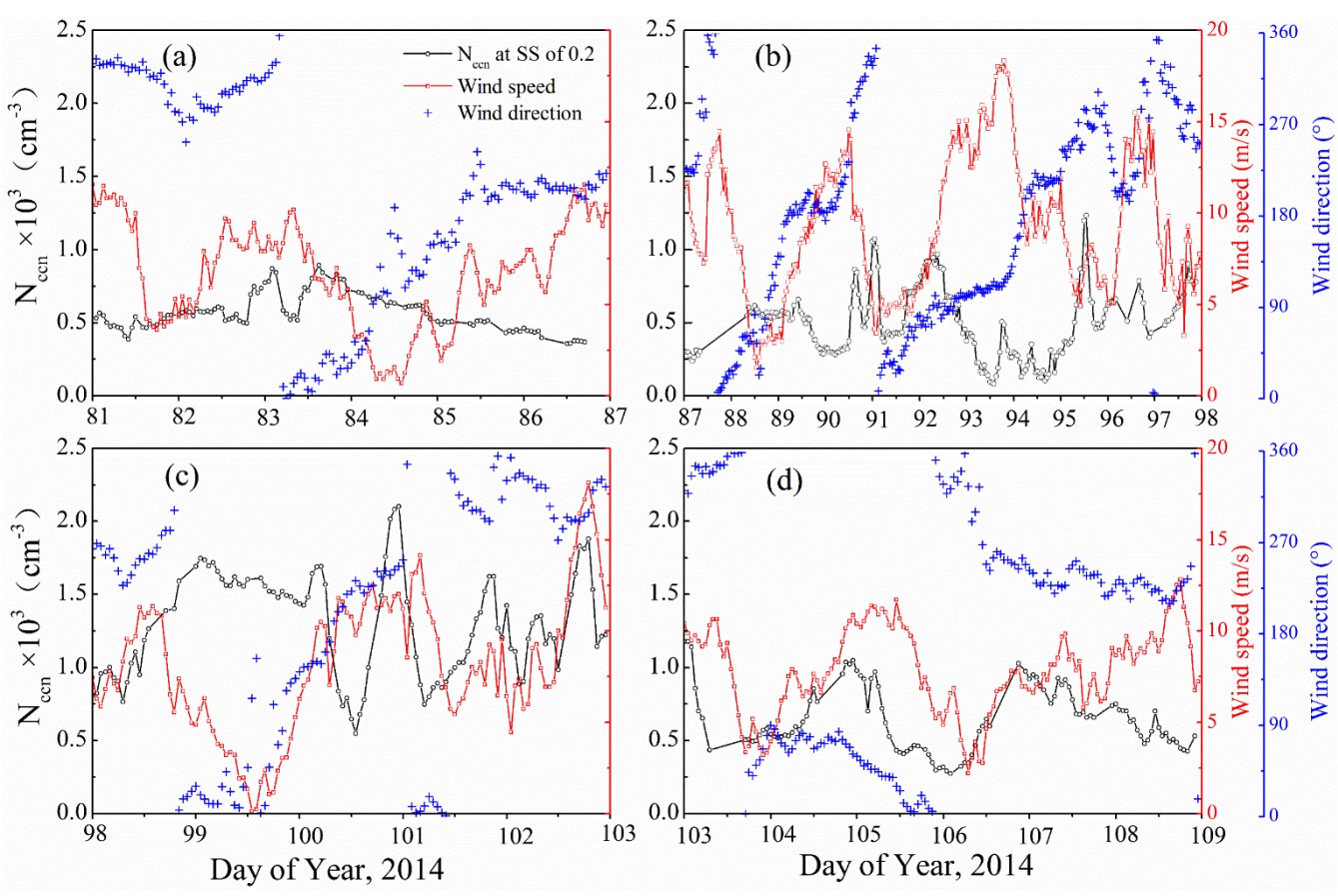

**Figure 6** Time series of the $N_{ccn}$, wind speed and wind direction at SS of 0.2% during the measurement for the four periods, including Period 1 (a), Period 2 (b), Period 3 (c) and Period 4 (d).

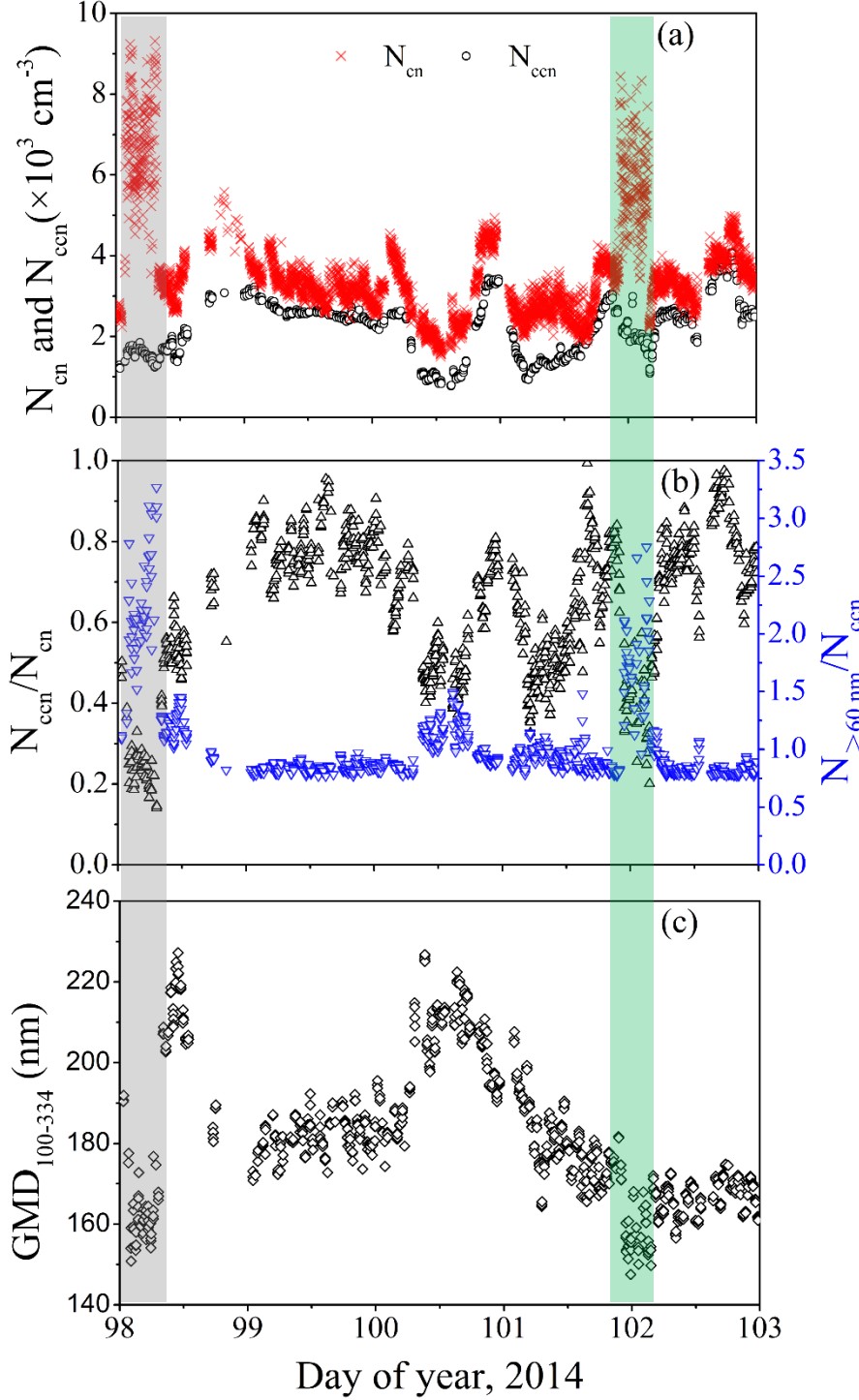

**Figure 7** Time series of the $N_{cn}$, the bulk $N_{ccn}$, $N_{ccn}/N_{cn}$, $N_{>60\,nm}/N_{ccn}$ and GMD $_{100\text{-}334}$ at SS of 0.4% during DOY 98-102, 2014. The gray shading indicates the dust event and the green shading indicates the BB event.

**Table 1** Concentrations of CN and CCN, AR, mixing ratios of $O_3$ and metrological conditions during DOY 81-108, 2014 over the NWPO.

| Sampling periods | DOY 81-86 (Period 1) | DOY 87-97 (Period 2) | DOY 98-102 (Period 3) | DOY 103-108 (Period 4) | DOY 81-108 |
|---|---|---|---|---|---|
| CN ($\times10^3$) [a] | 1.4-4.9, 2.8±0.51 [b] | 1.7-5.0, 3.1 ± 0.61* | 1.5-9.3, 3.5±1.2 | 0.88-3.8, 2.0±0.53 | 0.88-9.3, 2.8±1.0 |
| CCN($\times10^3$), SS=0.2% | 0.34-0.94 | 0.07-1.4 | 0.53-2.2 | 0.25-1.3 | 0.07-2.2 |
| | 0.56±0.12 | 0.50±0.24 | 1.3±0.36 | 0.64±0.21 | 0.68±0.38 |
| CCN($\times10^3$), SS=0.4% | 0.47-1.6 | 0.12-2.3 | 0.77-4.1 | 0.33-2.6 | 0.12-4.1 |
| | 0.84±0.20 | 0.79±0.38 | 2.2±0.72 | 1.0±0.41 | 1.1±0.67 |
| AR, 0.2%SS | 0.09-0.38 | 0.11-0.42 | 0.09-0.65 | 0.13-0.59 | 0.08-0.65 |
| | 0.21±0.06 | 0.21±0.06* | 0.38±0.11 | 0.32±0.08 | 0.30±0.11 |
| AR, 0.4%SS | 0.13-0.69 | 0.11-0.52 | 0.14-0.99 | 0.17-0.96 | 0.11-0.99 |
| | 0.32±0.09 | 0.30±0.08* | 0.64±0.18 | 0.49±0.13 | 0.46±0.19 |
| $O_3$ (ppb) | 7-73, 52±12 | 0-68, 37±17 | 12-74, 55±12 | 27-67, 49±9 | 0-74, 47±14 |
| RH (%) | 43-95, 62±13 | 56-100, 83±13 | 39-94, 66±15 | 45-98, 66±14 | 39-100, 70±15 |
| T (°C) | 13.2-20.5, 17.3±1.8 | 12.1-21.7, 18.2±1.7 | 10.5-19.6, 16.0±2.1 | 8.6-20.3, 15.9±3.3 | 8.6-22.5,17.1±2.8 |
| WS (m s$^{-1}$) | 0.9-11.6, 6.7±2.8 | 3.5-18.3, 10.1±4.1 | 0.8-18.3, 8.3±3.7 | 2.5-13.3, 7.9±2.3 | 0.8-18.3, 8.3±3.5 |

[a] Unit in $\times 10^3$ cm$^{-3}$

[b] Range and mean ± standard deviation

* The mean value during the measurement on DOY 96-97

**Table 2** Concentrations of CCN reported from different marine atmospheres.

| Sampling sites | Sampling period | $N_{ccn}$ (cm$^{-3}$) | Reference |
|---|---|---|---|
| NWPO | DOY 81-108, 2014 | 680 ± 380, SS = 0.2%; 1100 ± 670, SS = 0.4% | This study |
| Northwestern Pacific Ocean | Dec. 1994- Oct., 1996 | 165 ± 10, SS = 0.5%; 326 ± 18, SS = 1.0% (Continental) | Nagao et al. (1999) |
| | | 83 ± 16, SS = 0.5%; 146 ± 24, SS = 1.0% (Maritime) | |
| Southwest islands of Japan | 16 to 28 Apr., 2001 | 800-2000, SS = 0.3% | Adhikari et al. (2003) |
| Western North Pacific | Aug. to Sep., 2008 | <200, SS = 0.2%; <300, SS = 0.4%; Marine | Mochida et al. (2011) |
| | | <100, SS = 0.2%; ~100, SS = 0.4%; Volcanic plumes | |
| Eastern Pacific Ocean | Apr., 2004 | <300, SS = 0.2; <500, SS=0.4; | Roberts et al. (2006) |
| South China Sea | Sep. in both 2011 and 2012 | 320 ± 148, SS = 0.38%; Background | Atwood et al. (2017) |
| | Sep. in both 2011 and 2012 | 2340 ± 480, SS = 0.38%; Smoke | |
| | | 827 ± 270, SS = 0.38%; Mixed marine | |
| Southern Ocean | 17 Nov. to 12 Dec., 1995 | <100, SS = 0.2%; <300, SS = 1.0% | Hudson et al. (1998) |
| Arctic | 3 Aug. to 9 Sep., 2008 | <50, SS = 0.2%; <100, SS = 0.4% | Leck and Svensson. (2015) |
| Western North Atlantic | mid-June to mid-July 2013 | 173, SS = 0.2%; 205, SS = 0.4% | Kristensen et al. (2016) |
| Eastern Mediterranean | Sep. to Oct., 2007 | ~1250, SS = 0.21%; ~1750, SS = 0.38% | Bougiatioti et al. (2009) |
| Bay of Bengal | Jul., 2012 | 603 ± 400, SS = 0.2%; 963 ± 793, SS =0.4% | Chate et al. (2017) |
| | Aug., 2012 | 660 ± 624, SS = 0.2%; 1113 ± 1300, SS = 0.4% | |