# Peer review of "Nucleation—mode particle pool and large increases in $N_{cn}$ and $N_{ccn}$ observed over the northwestern Pacific Ocean in the spring of 2014"

_Atmospheric Chemistry and Physics, 2018_

## Referee Comment (RC1) · Anonymous Referee #2 · 25 Jan 2019

General comments: The manuscript by Wang et al, "Large increases in Ncn and Nccn together with a nucleation-mode-particle pool over the northwestern Pacific Ocean in the spring of 2014" reports CCN and CN concentration of maritime aerosols. As the authors said, CCN concentration itself of Asian outflow has not reported as literatures current years. The main result doesn't necessarily surprise; high concentration of CCN over their observation region is predictable from knowledge by previous observation reports for coastal and leeward area around East Asia and models. However, their fundamental data and reports for some events are valuable as observation reports. On the other hand, I was confused by the manuscript because some important information for representativeness and characteristic of the observation data were unexplained or

added later. This structure and title might mislead the readers to different image from author's assertion before read all the manuscript. Also, some topics were seen as lacking in explanation because the other possibilities were considered insufficiently. I have some question and comments to clarify the manuscript.

Specific comments:

1) Introduction: Authors said "direct observational data of aerosol particles and CCN in number concentrations remain limited in the remote atmosphere over the NWPO and the last spring observation can be traced back to 1996" as the motivation of the study. This is almost true, but this explanation can give different image of no recent studies for CCN in the region because they did not reference the relating studies in their introduction. There are several studies for CCN properties based on observation in similar air mass conditions. I think that CCN concentration tends to be seen as no urgent information because CCN concentration according to various conditions of supersaturations and CN concentration can be modeled by using accurate kappa values. What is the advantage of the direct observation of CCN concentration? Please clarify specific original point and information added to previous knowledge of CCN in remote sea of the East Asia regions.

2) Sections 3.1 and 3.2: Although air mass of the observation period tended to be affected from continental outflow, air mass in same region could be affected from marine air according to meteorological condition. The adequacy and meaning of discussions of continental input and estimation of kappa value depends on air mass tendency of the observation. I think that information of air mass tendency (Figure S3) should be explained before (or with) these discussions.

3) Section 3.2: Duseck et al. (2006) evaluated the correlation of CCN concentration estimated using constant composition or size distribution with the observed CCN; their evaluated point was different from this study. Correlation (R) is unnecessary to be well because aerosol species can have variation. Although authors used a size of "good

correlation (best R?)", slope=1 should be treated as most important if the aim was estimation of kappa, for discussion of both 0.4%SS and 0.2%SS. Also, some studies pointed a possibility of biased condition of air mass to the result of good correlation at constant compositions. Air mass condition of the analyses is important information to read implications of good correlation in this study. I think that data BB event was exclude in the analysis should be pointed in the manuscript. (Also, what is a rule of "suspected either BB or dust aerosols"? LEVO concentration?)

4) Section 3.2: For high Nccn/N60 at low CCN concentration, only effect of BB and dust was pointed in this study. Did you consider the other possibilities? Low Nccn can be observed at low Ntotal, low activation ability or both. In the case of Ntotal, there are possibilities of effect of diluting, transport of clean air mass and scavenging process etc. Because scavenging process can preferentially remove aerosols having high ability as CCN, the high Nccn/N60 and low CCN concentration can be obsevd. How were the Ntotal and the meteorological conditions?

5) The observation was conducted over marine, but comparatively near the continent of East Asia. Authors also suggested effect of continental input strongly. Therefore, I think that their observation result is valuable as "aged" air mass of continental pollution (after a few days) than aerosols over remote marine. Did the CCN properties (concentration and ability) in this study have difference to that of coastal area in East Asia (upwind area) by previous CCN studies? (Schmale et al. (ACP, 18, 2018) compiled resent CCN studies including information of CCN concentration around East Asia, which may also be useful to compare to this study.)

6) Section 3.3: I was confused; which did they assume temporal change of same air mass or regional difference? Although this section discussed mainly change by Hoppel effect, the difference of number-size distribution can include but only not effect by atmospheric process but also difference of origin of air mass. In this manuscript, many "increase" and "decrease" was used (e.g. P6L30, P7L6, L8, L9 etc.), especially in this section. However, I think that those without temporal change should be replaced "be

high" etc. In addition, Fig.1 is difficult to understand temporal change and representa-tiveness (fraction to all period) of the size distribution. Also, in some case, averaged distribution of 2-mode distribution having different peaks can become 3(or 4)-mode dis-tribution. It would be better to add temporal variation of number-size distribution (e.g. to Fig. 2). This is also helpful to show the accuracy of their data screening.

7) Sections 3.4 and 3.5: I was interested in the discussions, but data base on their observations seemed to not be enough to support the hypnosis that air mass was affected from upper layer. Cannot O3 data be used in this discussion?

8) I felt that the title was not sound right. This study did not observe direct relations between increase in CCN and CN and nucleation-mode particles. Also, temporal in-crease of CCN and CN was not shown in this study.

Tequnical comment and minor issues

Figure 1: The spots of map can be seen as fixed point. If the data included that during moving of ship, please add the ship track. In addition, legend is unclear and confusable with data spots. Also the direction should change.

Section 2.1: Please clarify where the inlet set. Also, did the data considered particle loss in tube?

Section 2.3: Was the data using screening only FMPS? CN seemed to be no data in period 2.

Section 3.2: Accuracy of kappa estimation depends on size classification. Please show how many bins of the analyzed size range.

———————————————

---

## Referee Comment (RC2) · Anonymous Referee #3 · 25 Mar 2019

This paper presents potentially interesting information on particle number size distributions and concentrations as well as CCN concentrations in a marine boundary layer. However, before recommending the acceptance of this paper for publications, several issues need to be discussed in more detail and care. My main points in this regard are given below.

Section 1. The authors need to define the scientific aims of this study more clearly in the paper. It is not enough to say what is being studied in the paper. The sentence "Several new findings have..." on page 3 sounds strange here.

Section 2.1. Description of the methods is incomplete. Nothing is said about the de-

tection limits of gas-phase instruments, or the performance of any of the instruments during the campaign. Was it tested whether the instruments (e.g. CCNC) performed during the measurements? The authors mention a correction factor of 1.25 for the FMPS but do not explain where this factor comes from. What are the potential effects of uncertainties in FMPS measurements on the results discussed in this paper? (something is mentioned on page 6, lines 23-26, but probably more is needed).

Section 3.1, last paragraph. There is a larger number of papers reporting CCN concentrations in the scientific literature. What was the basis for selecting these few studies when comparing results from this study? And why a single study conducted in Arctic was chosen here?

Section 3.3. After the more than 20-year-old papers citied here, a large number of studies (even reviews) on marine number size distributions have been published. The authors should make better use of these, more recent studies.

Section 3.4. Again, there are a number of more recent airborne studies on new particle formation in and above MFL in the scientific literature.

Section 3.5. What is the purpose of the two sentences on lines 27-29 in this section? Also the discussion at the end of this section is a bit confusing.

Technical issues:

Why do the authors use such complicated format when presenting concentrations (M+-N x 10-3). Would it be much simpler just to give the numbers as they are?

Page 6, line 11: Following those in the literature,. . . ????

---

## Author Comment (AC1) · 4 May 2019

General comments:

The manuscript by Wang et al, "Large increases in Ncn and Nccn together with a nucleation-mode-particle pool over the northwestern Pacific Ocean in the spring of 2014" reports CCN and CN concentration of maritime aerosols. As the authors said, CCN concentration itself of Asian outflow has not reported as literatures current years. The main result doesn't necessarily surprise; high concentration of CCN over their

observation region is predictable from knowledge by previous observation reports for coastal and leeward area around East Asia and models. However, their fundamental data and reports for some events are valuable as observation reports.

Response: The authors thank that the reviewer can agree the importance of direct observations of CN and CCN updated in the remote atmosphere over the NWPO. This is no doubt that modeling results are valuable to understand the climate effect of aerosols in the marine atmosphere. However, modeling results have to be constrained by direct observations of CCN. Upon this point, the observations delivered in this study are critical for accurately modeling Ncn and Nccn and potential climate effects of aerosol particles over the NWPO.

The observations in continental and marine atmospheres upwind the NWPO before 2010, e.g., Kim et al. (2014) and Adhikari et al. (2005) are important references. In the revision, a comprehensive comparison with the previous observations has been added to illustrate the characteristics of Ncn and Nccn over the NWPO in 2014.

On the other hand, I was confused by the manuscript because some important information for representativeness and characteristic of the observation data were unexplained or added later.

Response: In the revision, the authors try the best to address the concerns raised by the reviewer.

This structure and title might mislead the readers to different image from author's assertion before read all the manuscript.

Response: The title has been revised as "Nucleation-mode-particle pool and large increase in Ncn and Nccn observed over the northwestern Pacific Ocean in the spring of 2014"

Also, some topics were seen as lacking in explanation because the other possibilities were considered insufficiently. I have some question and comments to clarify the

manuscript.

Response: Please see our responses to the reviewer's specific comments.

Specific comments:

1) Introduction: Authors said "direct observational data of aerosol particles and CCN in number concentrations remain limited in the remote atmosphere over the NWPO and the last spring observation can be traced back to 1996" as the motivation of the study. This is almost true, but this explanation can give different image of no recent studies for CCN in the region because they did not reference the relating studies in their introduction.

Response: In spring, the NWPO receives a large amount of aerosol particles carried by the East Asian Monsoon. It is an ideal season to study the influence of Asian outflow on CN and CCN in the atmosphere over the NWPO. To best our knowledge, we don't find any observations of CN and CCN in the atmosphere over the NWPO in spring season after 1996. However, there was a large increase of air pollutants in emissions from upwind continents in the last two decades. We note that a few measurements of CN and CCN in the atmospheres upwind the NWPO are available after 1996 and the references have been added for a comprehensive comparison in the revision.

In summer, the East Asian Monsoon determines the NWPO to be less affected by continental air masses. As presented in the origin version, Mochida et al. (2011) made three-week measurements of CN and hygroscopic properties of aerosol in summer in 2008. We don't find other observations after this. However, we find more recent measurements of CN and CCN in other remote marine atmospheres because the types of data are still very limited world widely. The references have been cited and included in a comprehensive comparison.

There are several studies for CCN properties based on observation in similar air mass conditions. I think that CCN concentration tends to be seen as no urgent information

because CCN concentration according to various conditions of supersaturations and CN concentration can be modeled by using accurate kappa values. What is the advantage of the direct observation of CCN concentration? Please clarify specific original point and information added to previous knowledge of CCN in remote sea of the East Asia regions.

Response: In a recent research article published in Science, Rosenfeld et al. (2019) reported that lack of reliable estimates of CCN over oceans has severely limited our ability to quantify their effects on cloud properties and extent of cooling by reflecting solar radiation – a key uncertainty in anthropogenic climate forcing. Based on the article and the short comment by Sato and Suzuki (2019), it is safety to say that the previously estimated CCN in the marine atmosphere suffers from a larger error. Moreover, the CCN newly estimated by Rosenfeld et al. (2019), of course, still needs to be constrained by direct observations for warranting their accuracy. The references have been added in the revision.

The authors appreciate that the reviewer agrees direct observations of particle number size distributions to be needed for accurately estimating CCN. Our updated study has no doubt to fill the data scarcity. Regarding the aging processing of atmospheric aerosols, the authors cannot agree that those kappa values measured in upwind continental atmospheres can be used directly in the remote marine atmospheres. The same can be said that the measured kappa values in the summer clean marine atmosphere cannot be used directly in the spring marine atmosphere. For example, Wex et al. (2010) reported that the kappa values varied largely in marine atmospheres against those in continental atmospheres. Again, our direct observational data of Nccn can help more accurately evaluate the influence of Asian outflow of aerosols on the climate over the NWPO.

2) Sections 3.1 and 3.2: Although air mass of the observation period tended to be affected from continental outflow, air mass in same region could be affected from marine air according to meteorological condition. The adequacy and meaning of discussions

of continental input and estimation of kappa value depends on air mass tendency of the observation. I think that information of air mass tendency (Figure S3) should be explained before (or with) these discussions.

Response: We agree with the comments. The air mass back-trajectories have been presented in the origin version. In revised vision, more discussion has been added to analyze the origin of aerosol particles.

3) Section 3.2: Duseck et al. (2006) evaluated the correlation of CCN concentration estimated using constant composition or size distribution with the observed CCN; their evaluated point was different from this study. Correlation (R) is unnecessary to be well because aerosol species can have variation. Although authors used a size of "good correlation (best R?)", slope=1 should be treated as most important if the aim was estimation of kappa, for discussion of both 0.4%SS and 0.2%SS. Also, some studies pointed a possibility of biased condition of air mass to the result of good correlation at constant compositions. Air mass condition of the analyses is important information to read implications of good correlation in this study.

Response: The authors are very sorry since we cannot understand the comments. However, we revise the sentence as "As proposed in previous studies, e.g., Dusek et al. (2006) and Kalivitis et al. (2015), the total number concentration (N>Dp) of particles larger than a threshold diameter (Dp) can be used as a proxy for the Nccn. Specifically, aerosol particles with the size exceeding 60~70 nm could be activated as CCN at SS of 0.4% (Dusek et al., 2006)." We carefully check the paragraph in our manuscript and don't find others needed to be corrected.

Regarding Figs. 1, 2 and Table. 1 in Dusek et al. (2006), the aerosol particles with different air mass origins with the size exceeding 60~70 nm could be activated as CCN at SS of 0.4%. We thereby conducted regression analysis of the Nccn measured at SS of 0.4% against the N>Dp with Dp varying from 50 nm to 80 nm. We obtained the critical Dp to meet the slope of regression curve close to unity together with a good

correlation. Again, we strongly believe that our approach is valid.

I think that data BB event was exclude in the analysis should be pointed in the manuscript. (Also, what is a rule of "suspected either BB or dust aerosols"? LEVO concentration?)

Response: Agree. In the revision, we added "Note that data of biomass burning and dust aerosols and suspected either BB or dust aerosols were excluded in the analysis."

The values of N>60 nm/ Nccn were larger than 2.3 and 1.8 under dust and BB events, respectively. The data points with N>60 nm/ Nccn greater than 1.5 were clearly deviated from the general trend. Thus, we used the N>60 nm/ Nccn beyond 1.5 as a threshold to screen out either BB or dust aerosols as well as suspected BB or dust aerosols. This has been clarified in the revision.

4) Section 3.2: For high Nccn/N60 at low CCN concentration, only effect of BB and dust was pointed in this study. Did you consider the other possibilities? Low Nccn can be observed at low Ntotal, low activation ability or both. In the case of Ntotal, there are possibilities of effect of diluting, transport of clean air mass and scavenging process etc. Because scavenging process can preferentially remove aerosols having high ability as CCN, the high Nccn/N60 and low CCN concentration can be observed. How were the Ntotal and the meteorological conditions?

Response: We used and discussed the ratio of "N60/Nccn" through the manuscript rather than the ratio of "Nccn/N60" claimed by the reviewer. The comments appear to be irrelevant to our study.

We did analyze the relationship of Ncn with wind speed and wind direction (Fig. 1 at the end of this response), but we didn't find any correlation. Thus, we didn't include the inconclusive results in the manuscript.

5) The observation was conducted over marine, but comparatively near the continent of East Asia. Authors also suggested effect of continental input strongly. Therefore, I think

that their observation result is valuable as "aged" air mass of continental pollution (after a few days) than aerosols over remote marine. Did the CCN properties (concentration and ability) in this study have difference to that of coastal area in East Asia (upwind area) by previous CCN studies? (Schmale et al. (ACP, 18, 2018) compiled resent CCN studies including information of CCN concentration around East Asia, which may also be useful to compare to this study.)

Response: Schmale et al. (2018) used the data measured in Asia presented in two papers, i.e., Iwamoto et al. (2016) and Kim et al. (2014). A comparison including these data has been added in the revision.

The reviewer commented that "I think that their observation result is valuable as "aged" air mass of continental pollution (after a few days) than aerosols over remote marine." The authors believe that the reviewer may mix two technical terms, i.e., ocean-derived aerosols and aerosols observed in the marine atmosphere.

6) Section 3.3: I was confused; which did they assume temporal change of same air mass or regional difference? Although this section discussed mainly change by Hoppel effect, the difference of number-size distribution can include but only not effect by atmospheric process but also difference of origin of air mass. In this manuscript, many "increase" and "decrease" was used (e.g. P6L30, P7L6, L8, L9 etc.), especially in this section. However, I think that those without temporal change should be replaced "be high" etc. In addition, Fig.1 is difficult to understand temporal change and representativeness (fraction to all period) of the size distribution. Also, in some case, averaged distribution of 2-mode distribution having different peaks can become 3(or 4)-mode distribution. It would be better to add temporal variation of number-size distribution (e.g. to Fig. 2). This is also helpful to show the accuracy of their data screening.

Response: Honestly, the authors don't fully capture what the reviewer was trying to say. Based on the authors' guess, the reviewer was arguing that temporal changes in particle number size distribution observed over the NWPO were unrealistic when the

air masses originated from the same continents upwind. The argument is clearly invalid because the same continents upwind the NWPO can experience various air pollution scenarios in different periods, e.g., a heavy pollution event, a moderately air pollution event, a clear air quality event, a dust event and a biomass burning event, etc. The same can be said for air masses originated from different continents, e.g., from the Siberia, the north China, Japan, etc.

The language has been edited by an English editor and we also don't find any misleading by using "decrease and increase" in the context.

We add the contour plotting of particle number size distribution through the whole cruise period as supporting information in the revision. We, however, strongly believe that the daily average with standard deviation is a reasonable choice to present our results. We prefer to keep Fig. 1 in the context. For daily average particle number concentration, it is not surprised to see a broad peak because of the changed number concentrations in different times.

7) Sections 3.4 and 3.5: I was interested in the discussions, but data base on their observations seemed to not be enough to support the hypnosis that air mass was affected from upper layer. Cannot O3 data be used in this discussion?

Response: Thank for reviewer's interest. Vertical backward air mass trajectories have been added in the revision. The 3-day back trajectories showed that air masses were transported mostly from Asian continent at high altitude (>3000 m a.m.s.l.) and then mixed downward to the atmosphere near the sea-level. The related analysis has been added accordingly. Unlike in the continental atmospheres, no clear diurnal variation of O3 can be observed in the marine atmospheres. Therefore, O3 is not a good indicator to study the vertical transport in the marine atmospheres.

8) I felt that the title was not sound right. This study did not observe direct relations between increase in CCN and CN and nucleation-mode particles. Also, temporal increase of CCN and CN was not shown in this study.

Response: We revised title as "Nucleation-mode-particle pool and large increase in Ncn and Nccn observed over the northwestern Pacific Ocean in the spring of 2014"

Tequnical comment and minor issues

Figure 1: The spots of map can be seen as fixed point. If the data included that during moving of ship, please add the ship track. In addition, legend is unclear and confusable with data spots. Also the direction should change.

Response: In order to avoid clustering, the ship track was not shown in Fig. 1. However, we provide the cruise track in supporting information (Fig. S1) in the revision. The legend resolution has been improved. However, the authors cannot adjust the legend format because it generates by the software automatically. The authors find that the legend format is common in literature.

Section 2.1: Please clarify where the inlet set. Also, did the data considered particle loss in tube?

Response: In the revised method section, it reads as "All instruments were placed in the lab at the sixth floor of the vessel and approximately 15 m above the sea level. Atmospheric particles were sampled through conductive tubes (TSI, US) connected with a diffusion dryer filled with silica gel (TSI, US) and a splitter that split the air flow into different instruments. The tube inlet was stretched out the window of the cabin linking to the bridge. The total sampling line is approximately 1.5 m and the loss for > 10 nm particles is tested to be negligible."

A series of experiments had been conducted to test particle loss in the tube in 1.5-meter length. The loss varied from undetectable to 8% with the average of 4%. Since the loss is much smaller than the analytic error of the instrument and we had no correction for the raw data on this point.

Section 2.3: Was the data using screening only FMPS? CN seemed to be no data in period 2.

Response: In the Section 2.3, we detailed on how to screen out the data. The Ncn during Period 2 was not available because of instrument malfunction.

Section 3.2: Accuracy of kappa estimation depends on size classification. Please show how many bins of the analyzed size range.

Response: The FMPS includes 32 bins to measure number particle size concentration, in which 19 size bins covers the size range below 100 nm and 13 bins cover the size range beyond 100 nm. This has been added in the revision.

References:

Adhikari, M., Ishizaka, Y., Minda, H., Kazaoka, R., Jensen, J. H., Gras, J. L.: Vertical distribution of cloud condensation nuclei concentrations and their effect on microphysical properties of clouds over the sea near the southwest islands of Japan. J. Geophys. Res., 110, D10203, doi: 10.1029/2004JD004758, 2005.

Dusek, U., Frank, G. P., Hildebrandt, L., Curtius, J., Schneider, J., Walter, S., Chand, D., Drewnick, F., Hings, S., Jung, D., Borrmann, S. and Andreae, M. O.: Size matters more than chemistry aerosol particles, Science, 312(5778), 1375–1378, doi:10.1126/science.1125261, 2006.

Iwamoto, Y., Kinouchi, K., Watanabe, K., Yamazaki, N. and Matsuki, A.: Simultaneous measurement of CCN Activity and chemical composition of fine-mode aerosols at Noto Peninsula, Japan, in autumn 2012, Aerosol Air Qual. Res., 16, 2107-2118, doi:10.4209/aaqr.2015.09.0545, 2016.

Kalivitis, N., Kerminen, V., Kouvarakis, G., Stavroulas, I., Bougiatioti, A., Nenes, A., Manninen, H., Petäjä, T., Kulmala, M., Mihalopoulos, N.: Atmospheric new particle formation as a source of CCN in the eastern Mediterranean marine boundary layer. Atmos. Chem. Phys., 15, 9203-9215, 2015.

Kim, J. H., Yum, S. S., Shim, S., Kim, W. J., Park, M., Kim, J. H., Kim, M. H. and Yoon, S. C.: On the submicron aerosol distributions and CCN number concentrations in and around the Korean Peninsula, Atmos. Chem. Phys., 14(16), 8763– 8779, doi:10.5194/acp-14-8763-2014, 2014.

Mochida, M., Nishita-Hara, C., Furutani, H., Miyazaki, Y., Jung, J., Kawamura, K. and Uematsu, M.: Hygroscopicity and cloud condensation nucleus activity of marine aerosol particles over the western North Pacific, J. Geophys. Res. Atmos., 20 116(6), 1–16, doi:10.1029/2010JD014759, 2011.

Rosenfeld, D., Zhu, Y., Wang, M., Zheng, Y., Goren, T., Yu, S.: Aerosol-driven droplet concentrations dominate coverage and water of oceanic low-level clouds. Science 363, eaav0566. DOI: 10.1126/science.aav0566, 2019.

Schmale, J., Henning, S., Decesari, S., Henzing, B., Keskinen, H., Sellegri, K., Ovadnevaite, J., Pöhlker, M., Brito, J., Bougiatioti, A., Kristensson, A., Kalivitis, A., Stavroulas, I., Carbone, S., Jefferson, A., Park, M., Schlag, P., Iwamoto, Y., Aalto, P., Äijälä, M., Bukowiecki, N., Ehn, M., Frank, G., Fröhlich, R., Frumau, A., Herrmann, E., Herrmann, H., Holzinger, R., Kos, G., Kulmala, M., Mihalopoulos, N., Nenes, A., Amp, O., Dowd, A., Petäjä, T., Picard, D., Pöhlker, C., Pöschl, U., Poulain, L., Prévôt, A., Swietlicki, E., Andreae, M., Artaxo, P., Wiedensohler, A., Ogren, J., Matsuki, A., Yum, S., Stratmann, F., Baltensperger, U., Gysel, M.: Long-term cloud condensation nuclei number concentration, particle number size distribution and chemical composition measurements at regionally representative observatories. Atmos. Chem. Phys., 18, 2853-2881, doi.org/10.5194/acp-18-2853-2018, 2018.

Sato, Y., Suzuki, K.: How do aerosols affect cloudiness? Science, 36 (6427), 580-581, doi: 10.1126/science.aaw3720,2019.

Wex, H., McFiggans, G., Henning, S., Stratmann F.: Influence of the external mixing state of atmospheric aerosol on derived CCN number concentrations. Geophys. Res. Lett., 37, L10805, doi:10.1029/2010GL043337, 2010.

Please also note the supplement to this comment:

[Figure]

https://www.atmos-chem-phys-discuss.net/acp-2018-1089/acp-2018-1089-AC1-supplement.pdf

[Figure]

[Figure]

**Fig. 1.** Time series of the Ncn, wind speed and wind direction during the measurement for the four periods, including Period 1 (a), Period 2 (b), Period 3 (c) and Period 4 (d).

---

## Author Comment (AC2) · 4 May 2019

General comments:

This paper presents potentially interesting information on particle number size distributions and concentrations as well as CCN concentrations in a marine boundary layer. However, before recommending the acceptance of this paper for publications, several issues need to be discussed in more detail and care. My main points in this regard are given below.

[Figure]

Response: Thanks for reviewer's comments. We try our best to revise the manuscript accordingly.

Section 1. The authors need to define the scientific aims of this study more clearly in the paper. It is not enough to say what is being studied in the paper. The sentence "Several new findings have. . ." on page 3 sounds strange here.

Response: Agree and revise accordingly. In the revision, page 2, lines 14-19, we add "Due to practical difficulties, direct observations of Nccn in the remote marine atmosphere are still limited and this restrains to gain reliable estimates of the Nccn over the oceans, leading to the results of aerosol-cloud interaction estimates suffering from a larger error (Rosenfeld et al., 2019; Sato and Suzuki, 2019).", Page 2, Lines 29-31, we add "Moreover, modeling studies show that the NWPO likely suffer from the largest increase in surface sea temperature and experience the largest increase in $CO_2$ sink under warming climate (John et al., 2015; Lauvset et al., 2017). This further demonstrates the importance to study Ncn and NCCN and related potential climate effects therein." Page 3, Lines 3-5, we add "Through a comprehensive comparison with those observations in literature, we illustrated the characteristic of Ncn and Nccn over the NWPO in 2014 and revealed the changes in Ncn and Nccn against the results measured two decades ago. In addition, the influences of dust and BB aerosols on Ncn and Nccn were also analyzed on the monthly time scale."

Section 2.1. Description of the methods is incomplete. Nothing is said about the detection limits of gas-phase instruments, or the performance of any of the instruments during the campaign. Was it tested whether the instruments (e.g. CCNC) performed during the measurements? The authors mention a correction factor of 1.25 for the FMPS but do not explain where this factor comes from. What are the potential effects of uncertainties in FMPS measurements on the results discussed in this paper? (something is mentioned on page 6, lines 23-26, but probably more is needed).

Response: We revise the method parts accordingly. The detection limits of gas monitors have been added in the revision. Based on our recent measurements made by an on-line ion chromatography in remote marine atmospheres, we reluctantly used the data measured by the gas monitors to characterize the background concentrations of gaseous pollutants therein. We only used the data to help screen out ship self-emission. This has been clarified in the revision. It is almost impossible to test the performance of instruments during the cruise, but the instruments were tested after the campaign.

According to side-by-side measurements between the FMPS and a CPC during several campaigns before and after, the empirical coefficient of 1.25 was obtained. A comparison result has been added in supporting information (Fig. S2). The particle size reported by FMPS suffers from the errors against the results measured by the scanning mobility particle sizer (SMPS) (Lee et al., 2013), but the errors can be reasonably corrected using the empirical correction procedure proposed by Zimmenrman et al. (2015) to obtain highly consistent results with SMPS. We thereby conducted the correction in this study. This has been added in the revision.

Section 3.1, last paragraph. There is a larger number of papers reporting CCN concentrations in the scientific literature. What was the basis for selecting these few studies when comparing results from this study? And why a single study conducted in Arctic was chosen here?

Response: Agree. We add a long discussion on the comparison among those measurements in various marine atmospheres in the revision. Please see our revised Section 3.1.

Section 3.3. After the more than 20-year-old papers citied here, a large number of studies (even reviews) on marine number size distributions have been published. The authors should make better use of these, more recent studies.

Response: Agree. In the revision, we add "As reviewed by Vu et al. (2015), the particle number size distributions in the marine atmospheric boundary layer usually

showed two modes, such as Aitken mode and accumulation mode, with nucleation mode to be observed occasionally (Koponen et al., 2002; Ueda et al., 2016; Zhu et al., 2019). For example, the particle size number concentrations exhibited a bimodal distribution with an Aitken mode ($\sim$ 50 nm) and an accumulation mode (150-180 nm) during the fall campaign over the western North Pacific in 2008 (Mochida et al., 2011). The bimodal distributions were also reported during a winter campaign over the tropical and subtropical Pacific Oceans from 2011 to 2012 (Ueda et al., 2016) and during a campaign over the western North Atlantic in June-July 2013 (Kristensen et al., 2016). However, the Aitken mode and the accumulation mode were sometimes overlapped in the particle number size spectra measured over marginal seas influenced by polluted air masses (Lin et al., 2007; Nair et al., 2013; Zhu et al., 2019)."

Section 3.4. Again, there are a number of more recent airborne studies on new particle formation in and above MFL in the scientific literature.

Response: In the revision, Page 9, bottom paragraph, we add "Recent measurements further supported that NPF events most likely occurred in the free troposphere over different oceanic zones (Dadashazar et al., 2018; Rose et al., 2015; Sanchez et al., 2018; Takegawa et al., 2014). Several factors such as the lower temperatures and lower relatively humility, lower condensation sinks, mixed precursors from the continent and marine sources lower in free troposphere have also been argued why new particle formation occurred therein."

In the revised last paragraph of Section 3.4, we add "Including the increase in Ncn by NPF events, a few studies proposed that the nucleation mode particles may grow to be larger, even reach CCN size in the free troposphere (Rose et al., 2015; Sanchez et al., 2018) and have size growth during subsidence process from free troposphere to MBL. Sanchez et al. (2018) estimated that the contributions of NPF in the free troposphere to the Nccn at SS of 0.1% in the clean marine atmosphere over the North Atlantic were 31% and 33% in late-autumn and late-spring, respectively. Merikanto et al. (2009) reported that 55% of CCN at SS of 0.2% in the marine boundary layer were

from nucleation, with 45% entrained from the free troposphere and reminding 10% nucleated directly in the boundary layer. However, the growth of newly formed particles to CCN size was not observed in this study."

Section 3.5. What is the purpose of the two sentences on lines 27-29 in this section? Also the discussion at the end of this section is a bit confusing.

Response: In revision, we add "Regarding of the vertical distribution of the Nccn over the marine atmosphere, three scenario are hypothesized: 1) the Nccn aloft was larger than those in the atmosphere near the sea level; 2) the Nccn in vertical direction was homogenous; 3) the Nccn aloft was lower than those in the atmosphere near the sea level. Varying wind speeds may change the convection, which in turn affected the Nccn in the atmosphere near the sea level. For example, Clarke et al. (2013) reported that CCN activated in MBL clouds were strongly influenced by entrainment from the free troposphere (FT). Zheng et al. (2018) also argued that entrainment of FT aerosols was a vital source of accumulation mode particles over the eastern North Atlantic, which could activate as CCN easily."

The last part has been revised as "Based on vertical backward air mass trajectories (Fig. S6), air masses were transported mostly from the Asian continent at high altitude (>2000 m a.m.s.l.) to the reception zones, indicating that air masses were affected by the entrainment of FT aerosols. Therefore, it is reasonable to argue that the Nccn mixed downward from FT may be an important source of the Nccn in the MBL over the NWPO. However, modeling studies are needed to quantify the contribution in the future."

Technical issues:

Why do the authors use such complicated format when presenting concentrations (M+-N x 10-3). Would it be much simpler just to give the numbers as they are?

Response: Considering analytic errors of FMPS and the effective number of number

concentrations to be consistent with analytic errors, we used this format to present our results.

Page 6, line 11: Following those in the literature,. . . ????

Response: The part has been revised as "As proposed in previous studies, e.g., Dusek et al. (2006) and Kalivitis et al. (2015), the total number concentration (N>Dp) of particles larger than a threshold diameter (Dp) can be used as a proxy for the Nccn. Specifically, aerosol particles with the size exceeding 60~70 nm could be activated as CCN at SS of 0.4% (Dusek et al., 2018). In this study, N>Dp with Dp varying from 50 nm to 80 nm was calculated and a linear correlation was conducted with the values of Nccn measured at SS of 0.4%. A good correlation was obtained between Nccn and N>60 nm, with the slope of 0.98 closer to unity and R2=0.94 (Fig. 3)."

References:

[revised manuscript text omitted]

Please also note the supplement to this comment:
https://www.atmos-chem-phys-discuss.net/acp-2018-1089/acp-2018-1089-AC2-supplement.pdf

---

## Author Response (AR1)

*General comments:*

*The manuscript by Wang et al, "Large increases in $N_{cn}$ and $N_{ccn}$ together with a nucleation-mode-particle pool over the northwestern Pacific Ocean in the spring of 2014" reports CCN and CN concentration of maritime aerosols. As the authors said, CCN concentration itself of Asian outflow has not reported as literatures current years. The main result doesn't necessarily surprise; high concentration of CCN over their observation region is predictable from knowledge by previous observation reports for coastal and leeward area around East Asia and models. However, their fundamental data and reports for some events are valuable as observation reports.*

**Response:** The authors thank that the reviewer can agree the importance of direct observations of CN and CCN updated in the remote atmosphere over the NWPO. This is no doubt that modelling results are valuable to understand the climate effect of aerosols in the marine atmosphere. However, modelling results have to be constrained by direct observations of CCN. Upon this point, the observations delivered in this study are critical for accurately modelling $N_{cn}$ and $N_{ccn}$ and potential climate effects of aerosol particles over the NWPO.

The observations in continental and marine atmospheres upwind the NWPO before 2010, e.g., Kim et al. (2014) and Adhikari et al. (2005) are important references. In the revision, a comprehensive comparison with the previous observations has been added to illustrate the characteristics of $N_{cn}$ and $N_{ccn}$ over the NWPO in 2014.

*On the other hand, I was confused by the manuscript because some important information for representativeness and characteristic of the observation data were unexplained or added later.*

**Response:** In the revision, the authors try the best to address the concerns raised by the reviewer.

*This structure and title might mislead the readers to different image from author's assertion before read all the manuscript.*

**Response:** The title has been revised as "Nucleation–mode particle pool and large increases in $N_{cn}$ and $N_{ccn}$ observed over the northwestern Pacific Ocean in the spring of 2014"

*Also, some topics were seen as lacking in explanation because the other possibilities were considered insufficiently. I have some question and comments to clarify the manuscript.*

**Response:** Please see our responses to the reviewer's specific comments.

*Specific comments:*

*1) Introduction: Authors said "direct observational data of aerosol particles and CCN in number concentrations remain limited in the remote atmosphere over the NWPO and the last spring observation can be traced back to 1996" as the motivation of the study. This is almost true, but this explanation can give different image of no recent studies for CCN in the region because they did not reference the relating studies in their introduction.*

**Response:** In spring, the NWPO receives a large amount of aerosol particles carried by the East Asian Monsoon. It is an ideal season to study the influence of Asian outflow on CN and CCN in the atmosphere over the NWPO. To best our knowledge, we don't find any observations of CN and CCN in the atmosphere over the NWPO in spring season after 1996. However, there was a large increase of air pollutants in emissions from upwind continents in the last two decades. We note that a few measurements of CN and CCN in the atmospheres upwind the NWPO are available after 1996 and the references have been added for a comprehensive comparison in the revision.

In summer, the East Asian Monsoon determines the NWPO to be less affected by continental air masses. As presented in the origin version, Mochida et al. (2011) made three-week measurements of CN and hygroscopic properties of aerosol in summer in 2008. We don't find other observations after this. However, we find more recent measurements of CN and CCN in other remote marine atmospheres because the types of data are still very limited world widely. The references have been cited and included in a comprehensive comparison.

*There are several studies for CCN properties based on observation in similar air mass conditions. I think that CCN concentration tends to be seen as no urgent information because CCN concentration according to various conditions of supersaturations and CN concentration can be modeled by using accurate kappa values. What is the advantage of the direct observation of CCN concentration? Please clarify specific original point and information added to previous knowledge of CCN in remote sea of the East Asia regions.*

**Response:** In a recent research article published in Science, Rosenfeld et al. (2019) reported that lack of reliable estimates of CCN over oceans has severely limited our ability to quantify their effects on cloud properties and extent of cooling by reflecting solar radiation – a key uncertainty in anthropogenic climate forcing. Based on the article and the short comment by Sato and Suzuki (2019), it is safety to say that the previously estimated CCN in the marine atmosphere suffers from a larger error. Moreover, the CCN newly estimated by Rosenfeld et al. (2019), of course, still needs to be constrained by direct observations for warranting their accuracy. The references have been added in the revision.

The authors appreciate that the reviewer agrees direct observations of particle number size distributions to be needed for accurately estimating CCN. Our updated study has no doubt to fill the data scarcity. Regarding the aging processing of atmospheric aerosols, the authors cannot agree that those *kappa* values measured in upwind continental atmospheres can be used directly in the remote marine atmospheres. The same can be said that the measured kappa values in the summer clean marine atmosphere cannot be used directly in the spring marine atmosphere. For example, Wex et al. (2010) reported that the *kappa* values

varied largely in marine atmospheres against those in continental atmospheres. Again, our direct observational data of $N_{ccn}$ can help more accurately evaluate the influence of Asian outflow of aerosols on the climate over the NWPO.

*2) Sections 3.1 and 3.2: Although air mass of the observation period tended to be affected from continental outflow, air mass in same region could be affected from marine air according to meteorological condition. The adequacy and meaning of discussions of continental input and estimation of kappa value depends on air mass tendency of the observation. I think that information of air mass tendency (Figure S3) should be explained before (or with) these discussions.*

**Response:** We agree with the comments. The air mass back-trajectories have been presented in the origin version. In revised vision, more discussion has been added to analyze the origin of aerosol particles.

*3) Section 3.2: Duseck et al. (2006) evaluated the correlation of CCN concentration estimated using constant composition or size distribution with the observed CCN; their evaluated point was different from this study. Correlation (R) is unnecessary to be well because aerosol species can have variation. Although authors used a size of "good correlation (best R?)", slope=1 should be treated as most important if the aim was estimation of kappa, for discussion of both 0.4%SS and 0.2%SS. Also, some studies pointed a possibility of biased condition of air mass to the result of good correlation at constant compositions. Air mass condition of the analyses is important information to read implications of good correlation in this study.*

**Response:** The authors are very sorry since we cannot understand the comments. However, we revise the sentence as "As proposed in previous studies, e.g., Dusek et al. (2006) and Kalivitis et al. (2015), the total number concentration ($N_{>Dp}$) of particles larger than a threshold diameter ($D_p$) can be used as a proxy for $N_{ccn}$. Specifically, aerosol particles with sizes exceeding 60–70 nm could be activated as CCN at an SS of 0.4% (Dusek et al., 2006)." We carefully check the paragraph in our manuscript and don't find others needed to be corrected.

Regarding Figs. 1, 2 and Table. 1 in Dusek et al. (2006), the aerosol particles with different air mass origins with the size exceeding 60~70 nm could be activated as CCN at SS of 0.4%. We thereby conducted regression analysis of the $N_{ccn}$ measured at SS of 0.4% against the $N_{>Dp}$ with $D_p$ varying from 50 nm to 80 nm. We obtained the critical $D_p$ to meet the slope of regression curve close to unity together with a good correlation. Again, we strongly believe that our approach is valid.

*I think that data BB event was exclude in the analysis should be pointed in the manuscript. (Also, what is a rule of "suspected either BB or dust aerosols"? LEVO concentration?)*

**Response:** Agree. In the revision, we added "Note that data of biomass burning and dust aerosols and suspected either BB or dust aerosols were excluded in the analysis."

The values of $N_{>60\,nm}/N_{ccn}$ were larger than 2.3 and 1.8 under dust and BB events, respectively. The data points with $N_{>60\,nm}/N_{ccn}$ greater than 1.5 were clearly deviated from the general trend. Thus, we used the $N_{>60\,nm}/N_{ccn}$ beyond 1.5 as a threshold to screen out either BB or dust aerosols as well as suspected BB or dust aerosols. This has been clarified in the revision.

*4) Section 3.2: For high Nccn/N60 at low CCN concentration, only effect of BB and dust was pointed in this study. Did you consider the other possibilities? Low Nccn can be observed at low Ntotal, low activation ability or both. In the case of $N_{total}$, there are possibilities of effect of diluting, transport of clean air mass and scavenging process etc. Because scavenging process can preferentially remove aerosols having high ability as CCN, the high Nccn/N60 and low CCN concentration can be observed. How were the Ntotal and the meteorological conditions?*

**Response:** We used and discussed the ratio of "$N_{60}/N_{ccn}$" through the manuscript rather than the ratio of "$N_{ccn}/N_{>60\,nm}$" claimed by the reviewer. The comments appear to be irrelevant to our study.

[Figure]

**Figure R1** Time series of the $N_{cn}$, wind speed and wind direction during the measurement for the four periods, including Period 1 (a), Period 2 (b), Period 3 (c) and Period 4 (d).

We did analyze the relationship of $N_{cn}$ with wind speed and wind direction (Fig. R1), but we didn't find any correlation. Thus, we didn't include the inconclusive results in the manuscript.

*5) The observation was conducted over marine, but comparatively near the continent of East Asia. Authors also suggested effect of continental input strongly. Therefore, I think that their observation result is valuable as "aged" air mass of continental pollution (after a few days) than aerosols over remote marine. Did the CCN properties (concentration and ability) in this study have difference to that of coastal*

*area in East Asia (upwind area) by previous CCN studies? (Schmale et al. (ACP, 18, 2018) compiled resent CCN studies including information of CCN concentration around East Asia, which may also be useful to compare to this study.)*

**Response:** Schmale et al. (2018) used the data measured in Asia presented in two papers, i.e., Iwamoto et al. (2016) and Kim et al. (2014). A comparison including these data has been added in the revision.

The reviewer commented that *"I think that their observation result is valuable as "aged" air mass of continental pollution (after a few days) than aerosols over remote marine."* The authors believe that the reviewer may mix two technical terms, i.e., ocean-derived aerosols and aerosols observed in the marine atmosphere.

*6) Section 3.3: I was confused; which did they assume temporal change of same air mass or regional difference? Although this section discussed mainly change by Hoppel effect, the difference of number-size distribution can include but only not effect by atmospheric process but also difference of origin of air mass. In this manuscript, many "increase" and "decrease" was used (e.g. P6L30, P7L6, L8, L9 etc.), especially in this section. However, I think that those without temporal change should be replaced "be high" etc. In addition, Fig.1 is difficult to understand temporal change and representativeness (fraction to all period) of the size distribution. Also, in some case, averaged distribution of 2-mode distribution having different peaks can become 3(or 4)-mode distribution. It would be better to add temporal variation of number-size distribution (e.g. to Fig. 2). This is also helpful to show the accuracy of their data screening.*

**Response:** Honestly, the authors don't fully capture what the reviewer was trying to say. Based on the authors' guess, the reviewer was arguing that temporal changes in particle number size distribution observed over the NWPO were unrealistic when the air masses originated from the same continents upwind. The argument is clearly invalid because the same continents upwind the NWPO can experience various air pollution scenarios in different periods, e.g., a heavy pollution event, a moderately air pollution

event, a clear air quality event, a dust event and a biomass burning event, etc. The same can be said for air masses originated from different continents, e.g., from the Siberia, the north China, Japan, etc.

The language has been edited by an English editor and we also don't find any misleading by using "decrease and increase" in the context.

We add the contour plotting of particle number size distribution through the whole cruise period as supporting information in the revision. We, however, strongly believe that the daily average with standard deviation is a reasonable choice to present our results. We prefer to keep Fig. 1 in the context. For daily average particle number concentration, it is not surprised to see a broad peak because of the changed number concentrations in different times.

*7) Sections 3.4 and 3.5: I was interested in the discussions, but data base on their observations seemed to not be enough to support the hypnosis that air mass was affected from upper layer. Cannot O3 data be used in this discussion?*

**Response:** Thank for reviewer's interest. Vertical backward air mass trajectories have been added in the revision. The 3-day back trajectories showed that air masses were transported mostly from Asian continent at high altitudes (>2000 m a.m.s.l.) and then mixed downward to the atmosphere near the sea-level. The related analysis has been added accordingly. Unlike in the continental atmospheres, no clear diurnal variation of $O_3$ can be observed in the marine atmospheres. Therefore, $O_3$ is not a good indicator to study the vertical transport in the marine atmospheres.

*8) I felt that the title was not sound right. This study did not observe direct relations between increase in CCN and CN and nucleation-mode particles. Also, temporal increase of CCN and CN was not shown in this study.*

**Response:** We revised title as "Nucleation–mode particle pool and large increases in $N_{cn}$ and $N_{ccn}$ observed over the northwestern Pacific Ocean in the spring of 2014"

*Tequnical comment and minor issues*

*Figure 1: The spots of map can be seen as fixed point. If the data included that during moving of ship, please add the ship track. In addition, legend is unclear and confusable with data spots. Also the direction should change.*

**Response:** In order to avoid clustering, the ship track was not shown in Fig. 1. However, we provide the cruise track in supporting information (Fig. S1) in the revision. The legend resolution has been improved. However, the authors cannot adjust the legend format because it generates by the software automatically. The authors find that the legend format is common in literature.

*Section 2.1: Please clarify where the inlet set. Also, did the data considered particle loss in tube?*

**Response:** In the revised method section, it reads as "All instruments were placed in the lab at the sixth floor of the vessel approximately 15 m above sea level. Atmospheric particles were sampled through conductive tubes (TSI, US) connected with a diffusion dryer filled with silica gel (TSI, US) and a splitter that split the air flow into different instruments. The tube inlet was stretched from the window of the cabin linking to the bridge. The total sampling line was approximately 1.5 m and the loss for particles > 10 nm was tested to be negligible."

A series of experiments had been conducted to test particle loss in the tube in 1.5-meter length. The loss varied from undetectable to 8% with the average of 4%. Since the loss is much smaller than the analytic error of the instrument and we had no correction for the raw data on this point.

*Section 2.3: Was the data using screening only FMPS? CN seemed to be no data in period 2.*

**Response:** In the Section 2.3, we detailed on how to screen out the data. The $N_{cn}$ during Period 2 was not available because of instrument malfunction.

*Section 3.2: Accuracy of kappa estimation depends on size classification. Please show how many bins of the analyzed size range.*

**Response:** The FMPS includes 32 bins to measure number particle size concentration, in which 19 size bins covers the size range below 100 nm and 13 bins cover the size range beyond 100 nm. This has been added in the revision.

**Response:** We revise the method parts accordingly. The detection limits of gas monitors have been added in the revision. Based on our recent measurements made by an on–line ion chromatography in remote marine atmospheres, we reluctantly used the data measured by the gas monitors to characterize the background concentrations of gaseous pollutants therein. We only used the data to help screen out ship's self-emission. This has been clarified in the revision. It is almost impossible to test the performance of instruments during the cruise, but the instruments were tested after the campaign.

According to side–by–side measurements between the FMPS and a CPC during several campaigns previous and subsequent, the empirical coefficient of 1.25 was obtained. A comparison result has been added in supporting information (Fig. S2). Although the particle size reported by the FMPS showed errors against the results measured by the scanning mobility particle sizer (SMPS) (Lee et al., 2013), the errors were reasonably corrected using the empirical correction procedure proposed by Zimmenrman et al. (2015) to obtain highly consistent results with SMPS. We thereby conducted the correction in this study. This has been added in the revision.

*Section 3.1, last paragraph. There is a larger number of papers reporting CCN concentrations in the scientific literature. What was the basis for selecting these few studies when comparing results from this study? And why a single study conducted in Arctic was chosen here?*

**Response:** Agree. We add a long discussion on the comparison among those measurements in various marine atmospheres in the revision. Please see our revised Section 3.1.

*Section 3.3. After the more than 20-year-old papers cited here, a large number of studies (even reviews) on marine number size distributions have been published. The authors should make better use of these, more recent studies.*

**Response:** Agree. In the revision, we add "As reported by Vu et al. (2015), the particle number size distributions in the marine atmospheric boundary layer usually show two modes, Aitken mode and accumulation mode, with a nucleation mode observed occasionally (Koponen et al., 2002; Ueda et al., 2016; Zhu et al., 2019). For example, the particle size number concentrations exhibited a bimodal distribution with an Aitken mode (~ 50 nm) and an accumulation mode (150–180 nm) during the fall campaign over the western North Pacific in 2008 (Mochida et al., 2011). Bimodal distributions were also reported during a winter campaign over the tropical and subtropical Pacific Oceans from 2011 to 2012 (Ueda et al., 2016) and during a campaign over the western North Atlantic in June–July 2013 (Kristensen et al., 2016). However, the Aitken mode and the accumulation mode were sometimes overlapped in the particle number size spectra measured over marginal seas influenced by polluted air masses (Lin et al., 2007; Nair et al., 2013; Zhu et al., 2019)."

*Section 3.4. Again, there are a number of more recent airborne studies on new particle formation in and above MFL in the scientific literature.*

**Response:** In the revision, Page 9, bottom paragraph, we add "Recent measurements further support that NPF events most likely occur in the FT over different oceanic zones (Dadashazar et al., 2018; Rose et al., 2015; Sanchez et al., 2018; Takegawa et al., 2014). Several factors such as the lower temperatures and lower relatively humility, lower condensation sinks, and mixed precursors from the continental and marine sources lower in the FT have also been offered as explanations for the NPF occurrence therein."

In the revised last paragraph of Section 3.4, we add "Regarding the increase in $N_{cn}$ by NPF events, a few studies have proposed that nucleation–mode particles can increase in size, even reaching the CCN size in the FT (Rose et al., 2015; Sanchez et al., 2018) with the growth occurring during the subsidence process from FT to the MBL. Sanchez et al. (2018) estimated that the contributions of NPF in the FT to the $N_{ccn}$ at an SS of 0.1% in the clean marine atmosphere over the North Atlantic are 31% and 33% in late autumn and late spring, respectively. Merikanto et al. (2009) reported that 55% of CCN at an SS of 0.2% in the MBL are from nucleation, with 45% entrained from the FT and the reminding 10% nucleated directly in the boundary layer. However, growth of newly formed particles to the CCN size was not observed in this study."

*Section 3.5. What is the purpose of the two sentences on lines 27-29 in this section? Also the discussion at the end of this section is a bit confusing.*

**Response:** In revision, we add "Regarding of the vertical distribution of the $N_{ccn}$ over the marine atmosphere, three scenarios are hypothesized: 1) the $N_{ccn}$ aloft was larger than that in the atmosphere near sea level; 2) the $N_{ccn}$ in the vertical direction was homogenous; 3) or the $N_{ccn}$ aloft was lower than that in the atmosphere near sea level. Varying wind speeds may change the convection, which in turn affects $N_{ccn}$ in the atmosphere near sea level. For example, Clarke et al. (2013) reported that CCN activated in MBL clouds is strongly influenced by entrainment from the FT. Zheng et al. (2018) also argued that entrainment of FT aerosols is a vital source of accumulation mode particles over the eastern North Atlantic, which could be easily activated as CCN."

The last part has been revised as "According to the vertical backward air mass trajectories (Fig. S7), the air masses are transported mostly from the Asian continent at high altitudes (>2000 m a.m.s.l.) to the reception zones, indicating that air masses are affected by the entrainment of FT aerosols. Therefore, it is reasonable to argue that the $N_{ccn}$ mixed downward from the FT may be an important source of $N_{ccn}$ in the MBL over the NWPO. However, modeling studies are needed in the future to quantify the contribution."

*Technical issues:*

*Why do the authors use such complicated format when presenting concentrations (M+- N x 10-3). Would it be much simpler just to give the numbers as they are?*

**Response:** Considering analytic errors of FMPS and the effective number of number concentrations to be consistent with analytic errors, we used this format to present our results.

*Page 6, line 11: Following those in the literature,. . . ????*

[revised manuscript text omitted]
| | Aug., 2012 | $660 \pm 624$, SS = 0.2%; $1113 \pm 1300$, SS = 0.4% | |